# Refining Language Models with Compositional Explanations

**Huihan Yao[1]  Ying Chen[2]  Qinyuan Ye[3]  Xisen Jin[3]  Xiang Ren[3]**
[1]Peking University  [2]Tsinghua University  [3]University of Southern California
yaohuihan@pku.edu.cn  chenying17@mails.tsinghua.edu.cn
{qinyuany,xisenjin,xiangren}@usc.edu

## Abstract

Pre-trained language models have been successful on text classification tasks, but are prone to learning spurious correlations from biased datasets, and are thus vulnerable when making inferences in a new domain. Prior work reveals such spurious patterns via post-hoc explanation algorithms which compute the importance of input features. Further, the model is regularized to align the importance scores with human knowledge, so that the unintended model behaviors are eliminated. However, such a regularization technique lacks flexibility and coverage, since only importance scores towards a pre-defined list of features are adjusted, while more complex human knowledge such as feature interaction and pattern generalization can hardly be incorporated. In this work, we propose to refine a learned language model for a target domain by collecting *human-provided compositional explanations* regarding observed biases. By parsing these explanations into executable logic rules, the human-specified refinement advice from a small set of explanations can be generalized to more training examples. We additionally introduce a regularization term allowing adjustments for both importance and interaction of features to better rectify model behavior. We demonstrate the effectiveness of the proposed approach on two text classification tasks by showing improved performance in target domain as well as improved model fairness after refinement[1].

## 1  Introduction

With recent advances in model architectures and pre-training techniques, neural language models [4, 31, 25] have achieved impressive results on a broad set of natural language processing (NLP) tasks, such as sentiment analysis and hate speech detection [3, 45]. However, when a source model (fine-tuned on some upstream dataset) is applied to a target domain with a different data distribution, the model may suffer from poor performance due to some spurious feature patterns learned from the upstream dataset [33, 44, 6]. Moreover, some spurious patterns may cause unintended biases in the downstream tasks, resulting in fairness and trust concerns about the model [21].

Prior work suggests that humans can identify such spurious patterns through examining the visualized "heat-map" (Fig. 1) produced by a post-hoc model explanation algorithm [13]. As a prominent example, feature attribution methods [42, 14, 19] interpret model prediction on an instance by assigning an importance (attribution) score to each input feature (or token, in the context of NLP tasks), which helps uncover an overemphasis or understatement of a specific feature (e.g., "*Sweden*" is overemphasized as indication of hate speech, as in Fig. 1). To alleviate these spurious patterns, recent attempts study model regularization methods that update models in a differentiable and incremental fashion, which looks to align feature attribution scores with the "intended" scores manually specified by human annotators [38, 35, 30]. For example, attribution scores on overemphasised, unintended tokens are decreased (to close to zero) through updating the model weights [21].

---

[1]Code and data are available at https://github.com/INK-USC/expl-refinement.

35th Conference on Neural Information Processing Systems (NeurIPS 2021).

Despite these initial successes, existing model regularization methods have *limited capacity* in conveying complex human feedback regarding spurious patterns, and *limited regularization strength* as the regularization term is enforced on only instances associated with human feedback (while the vast amount of unlabeled data is not leveraged) [38, 36, 30, 47]. To pinpoint a spurious pattern precisely, an annotator needs to describe it by composing multiple assertions regarding feature attribution and feature interaction (*e.g.*, "*to be a failure*" modifies "*Sweden*" in Fig. 1). However, previous work consider only the former [36, 30] and omit the latter when characterizing spurious patterns. Moreover, refining these data-hungry language models

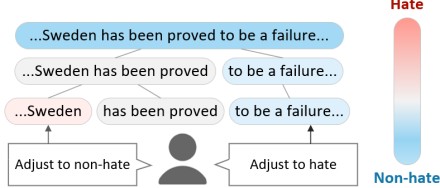

Figure 1: **An illustration of post-hoc model explanation heat-map for hate speech detection.** A trained hate speech classifier mis-classifies the sentence as non-hateful. After observing the heat-map, human annotators may suggest that "Sweden" be adjusted to neutral, and "failure" should contribute more to predicting hate speech.

often requires large amount of labeled data. To extend the coverage of regularization, one must match (generalize) one human feedback to multiple (unlabeled) instances in the target domain – *i.e.*, identify instances that potentially suffer from the same spurious patterns, and then regularize the model with a larger set of instances.

To this end, we introduce **Re**fining Language **Mo**del with Composi**t**ional **E**xplanation (REMOTE), a framework that alleviates spurious patterns of a trained model and addresses the aforementioned limitations, by soliciting complex and compositional explanations from human and refining the model with broadened coverage during regularization (see Fig. 2 for an overview). Firstly, human annotators are shown the post-hoc explanations of the source model's predictions on target domain data (Fig. 1). They are asked to describe the spurious patterns they find and their suggestions to adjust the importance scores and interactions of features. Secondly, we extract executable first-order logic rules from these human-provided compositional explanations. The execution of the logic rules are decomposed to several atomic operations. We devise softened versions of these operations so the logic rules provide noisy labels and refinement advice to a larger number of instances in the target domain. Lastly, we update model weights according to the suggestions in the explanations, using the enlarged set of instances obtained in the previous step.

We highlight two major contributions of the REMOTE framework. First, to the best of our knowledge, REMOTE is the first work that studies gathering feature-level supervision from complex human explanations. Prior work [15, 47] has explored producing pseudo labels from explanations (*i.e.*, what is the correct label?), while we focus on more concrete feature-level supervision (*i.e.*, why is this label correct?) with the goal of reducing spurious patterns. Second, we quantify and regularize the interaction between features, in addition to feature attributions used in prior work. This greatly improves the expressiveness of human explanations by supporting more complex rationale that involves more than one feature.

We validate our approach on three pairs of datasets in hate speech classification and sentiment analysis. Compared with direct transfer (evaluate the source model on target domain data) and other baselines (distillation and weight regularization), we observe notable performance improvements after refining the model with our proposed framework. In addition, we demonstrate that REMOTE can reduce unintended biases on group identifiers in hate speech detection.

## 2   Related Work

**Human-in-the-loop Learning.** The idea of bringing human into the learning process to enhance the model has been explored through multiple paths. One direction of prior work is to ask human annotators to label important instances (*i.e.* active learning [39]) or to determine which model to use [9]. A more interpretable direction is to associate features with specific labels as a source of supervision [32], or let users suggest adjustments of low-level features by examining model explanations [24, 43, 26]. These explanation-related methods are limited to relatively simple models including linear classifiers [24] and CNN [26] because the model explanations directly correspond to the low-level features. With model-agnostic post-hoc explanation algorithms that explain predictions of a trained model without interfering with its learned weights [42, 19], our interactive machine learning method enables inspection and feedback to models as complex as BERT. On the other hand, several works [15, 47, 51] argue that natural language explanations rather than numeric labels as human feedback provide a richer and more efficient source of supervision for training. Our method

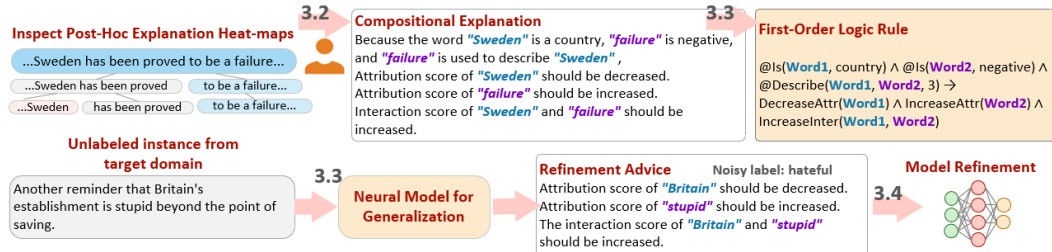

Figure 2: **Overview for model refinement with explanation regularization.** Post-hoc explanation heat-maps are presented to annotators, and compositional explanations describing the model refinement suggestions are collected (Sec. 3.2). We parse them to first-order rules and match unlabeled data in the target domain with a neural model (Sec. 3.3), and use them to refine the model (Sec. 3.4).

learns from natural language in different setting, where we aim to adapt a trained model to another domain. Another line of work has studied explanation regularization as an interpretable approach to impose prior knowledge into neural networks [38, 35, 36, 30, 7]. Compared with them, our work incorporate complicated human knowledge conveyed by natural language and further proposes regularization on feature interaction to better take context into consideration.

**Model Transfer and Domain Adaptation.** Sequential transfer learning, also known as *model transfer* [46], considers a model being trained sequentially over different labeled datasets that are not available at the same time. Existing model transfer methods update the model (using labeled data from target domain) with various fine-tuning techniques – *e.g.*, updating layer weights [8], creating learning rate schedule [18], adding regularization [48, 28, 27], and model distillation [17, 37]. More recent work also studies transferring models *without* labeled target data, known as *unsupervised model adaptation* [27], by generating target-like data [29], or learning target-specific features [27]. Our work goes beyond supervision in the form of "instance-label" data and considers more complex human feedback on feature attribution and interaction. In another relevant thread, *unsupervised domain adaptation (UDA)* looks to adapt a trained model to a new domain using unlabeled data from the target domain, by updating feature representation to minimize the distribution divergence between domains [12, 11], or updating models to match the distribution statistical moments at different orders [41, 52, 34]. However, UDA methods typically require access to labeled instances from the source domain, which are not available in our problem setting.

## 3 Model Refinement with Compositional Explanations

We study the problem of refining a *source model* for better adapting to a new domain (Sec. 3.1). By presenting the post-hoc *explanation heat-map* computed on the target data for the source model, we solicit from human annotators *compositional explanations* that describe what spurious patterns are observed in the instances and how to adjust the feature attribution and interaction scores to alleviate these spurious behaviors (Sec. 3.2). We aim to generalize the collected explanations to instances in the target domain (Sec. 3.3) and update model weights based on these explanation-generalized instances (Sec. 3.4).

### 3.1 Problem Formulation

We consider adapting a text classification model $f_S$ trained on some source data $\mathcal{D}_S^{train}$ (*e.g.*, a sentiment classifier trained on news articles) to a new *target domain* $T$ (*e.g.*, tweets) during a *model refinement* stage. We focus on a challenging setting [28, 29] where the upstream labeled data $\mathcal{D}_S^{train}$ is not available (*e.g.*, due to privacy constraints [27], or access restriction [29]) but unlabeled data in the target domain ($\mathcal{D}_T$) are readily accessible, during model refinement. This setting also reflects the common practice nowadays, as users may download trained models from public model repositories and look to deploy the model to their own data. Our problem setting is different from (unsupervised) domain adaptation [11, 34], where labeled source data ($\mathcal{D}_S^{train}$) is available for making model update. Our work also distinguishes from traditional transfer learning [18, 28] which focuses on leveraging labeled data in the form of "instance-label" pairs in the target domain [18], while we solicit human explanations on model's spurious patterns for adjusting feature attribution and interaction scores.

We evaluate model's performance on target data ($\mathcal{D}_T^{test}$) and source data ($\mathcal{D}_S^{test}$) as the measurement of success. We expect the target performance to be improved, while the source performance to be maximally preserved, after model refinement. In addition, a refined model should no longer rely on

spurious patterns (*e.g.*, being over-sensitive to group identifiers for hate speech detection). Therefore, we report False Positive Rate Difference (FPRD) on a synthetic dataset [5] as a fairness metric.

## 3.2 Compositional Explanation for Model Refinement

In the following, we define the format of compositional explanations used in our study, and the procedure to solicit these explanations from annotators.

**Compositional Explanations.** Our compositional explanations consist of two parts: *spurious patterns* and *refinement advice*. We define a *spurious pattern* as words or phrases in a sentence, whose attribution/interaction scores do *not* align with human judgment. Annotators are required to describe a spurious pattern precisely and sufficiently using three types of expressions: (1) existence of a feature; (2) characteristic of a feature; (3) relation between features. We list examples for these expressions in Table 1. Annotators then provide *refinement advice*, *i.e.*, their advice to increase or decrease the attribution/interaction scores of features. The annotator also provide a label for

Table 1: **Three types of expressions for describing spurious patterns.**

| |
| --- |
| **1. Existence of a feature** |
| **Example:** X is "jews". Y is "parasite". |
| **2. Characteristic of a single feature** |
| **Description:** Entity type, part-of-speech tag, sentiment label of a word/phrase. |
| **Example:** X is a Person entity. X is verb. Y is a positive word. |
| **3. Relation between features** |
| **Description:** Semantic roles, co-reference, distance, etc. |
| **Example:** X is the subject of Y. X is two tokens away from Y. X modifies Y. |

the current instance. Compositionality in human explanations can not only improve the precision of matching instances, but enable annotators to describe their observations more flexibly as well. We provide concrete examples of the compositional explanations in Table 2 and "Compositional Explanation" block in Fig. 2.

**Explanation Solicitation.** We first use the source model $f_S$ to make predictions on a small set of unlabeled instances randomly sampled from $\mathcal{D}_T$ and present these post-hoc explanation heat-maps [19] to human annotators (see Fig. 1). When shown with an instance with its heat-map, human annotators will: 1) read the sentence and provide a label, 2) inspect the heat-map, and 3) write an explanation if a spurious pattern is found. We estimate the time cost of each step and the ratio of finding spurious patterns, and will elaborate it in Sec. 4.2. If the annotator identifies a spurious pattern in the heat-map for instance $x_{ref}$ and provides an explanation, we refer to $x_{ref}$ as the *reference instance* for the explanation. In this step, we have obtained a set of instances $x_{ref}$ along with their raw human-provided explanations $e$, which we denote as $\mathcal{E}_0 = \{(x_{ref}, e)\}$.

## 3.3 Generalizing Explanation in Target Domain

With the collected human explanations, now we detail how to parse the collected natural-language explanations into executable logic rules $B \rightarrow H$, and how to execute those logic rules by softening their constraints so that they can be generalized to multiple unlabeled instances in $\mathcal{D}_T$.

**Explanation Parsing.** Each raw explanation $e$ is parsed into a first-order logic rule in the form of $B \rightarrow H$. Here the description of the spurious pattern is parsed into rule body $B$ and the refinement advice is parsed into rule head $H$ (see "First-Order Logic Rule" in Fig. 2). We parse the raw explanations using a semantic parser based on Combinatory Categorial Grammar (CCG) [53], which can deal with natural language with linguistic variations and thus is friendly to annotators. To tailor the parser to our needs, we define a lexicon $str2predicate$ to map 301 common expressions (*e.g.*, "directly after", "negative word") to 83 predicates and implement the corresponding operations with atomic modules (described in next paragraph). Annotators can iteratively modify their explanations or update the lexicon until they make sure their explanations are accepted by the parser. We denote the collection of parsed explanations as $\mathcal{E} = \{(x_{ref}, B, H)\}$. More details are in Appendix B.

**Explanation Generalization with Matching Model G.** With $\mathcal{E}$ obtained in previous steps, we now aim at generalizing from one explanation $(x_{ref}, B, H)$ to multiple unlabeled instances in $\mathcal{D}_T$. That is, for each unlabeled instance $x$ in $\mathcal{D}_T$, we attempt to match it with each spurious pattern $B$ and the reference instance $x_{ref}$ we have obtained. If the matching is successful, we consider that $x$ may suffer from a spurious pattern similar to $B$, and will use this instance in the later regularization phase.

For this purpose, we construct an executable matching model **G** from each rule body $B$. **G** is dynamically constructed from three atomic execution units (*Individuality Module*, *Interaction Module* and *Compositionality Module*), following the predicates in $B$. We introduce these modules as follows.

Individuality module is used to output the feature (*i.e.*, word) in the unlabeled instance $x$ that corresponds to a feature $q_{ref}$ in the reference instance $x_{ref}$. The module will first search if an exact match of $q_{ref}$ exists in $x_k$. If an exact match does not exist, it will search for words of same type with $q_{ref}$ in $x_k$, including named entity type, constituency parse structure, etc. We leave more details in Appendix E. If no feature is found after this step, Individuality module will return None.

Interaction module examines whether the relation between features described in $B$ holds true in the unlabeled instance $x_{ref}$. Given $x_{ref}$ and two words $q_{ref,1}, q_{ref,2}$ in it, we first call Individuality module to find matched $q_{k,1}, q_{k,2}$ in $x_k$, and then use Interaction module to examine their relation. We use either natural distance (number of words between the two words $q_{ref,1}, q_{ref,2}$) or dependency distance (distance between $q_{ref,1}, q_{ref,2}$ on the dependency tree), depending on the descriptions given by annotators. The former is applicable to explanations such as *"X is within 3 words before Y"*, and the latter is applicable to explanations such as *"Feature X modifies feature Y"*. The Interaction module will output True of False, indicating whether the relation described in $B$ is satisfied in $x_{ref}$.

Table 2: **An example of compositional explanation.** The explanation characterizes *where* a spurious pattern is (in the instance), and *how* the attribution/interaction scores should be regularized.

---
**Information shown to Annotators:**
**Reference Instance:** They prove more *distressing than attractive*.
**Information from Heat-map:** Predicted label is positive. Attribution score of *distressing* is low.
**Compositional Explanation**
**Spurious Pattern**: X is "*distressing*". Y is "*than*". Z is "*attractive*". X is negative. Z is positive. X is immediately before Y. Y is immediately before Z.
**Noisy Label:** Negative.
**Refinement Advice**: Attribution score of X should be increased.
**Matched Instance**
**Instance:** Self-flagellation is more *depressing than entertaining*.
**Noisy Label:** Negative.
**Refinement Advice:** X is "*depressing*". Attribution score of X should be increased. **Soft-Matching Details:** "distressing" and "depressing" are synonyms. "attractive" and "entertaining" are synonyms. The semantic similarity is captured by softened Individuality module.

---

Compositionality module is used to execute logic operations (*i.e.*, "AND", "OR") specified in the explanation. Based on intermediate output produced by other modules, the module will output True or False, indicating whether $B$ matches with an unlabeled instance $x$.

**Broadening Coverage with Softened Matching.** The matching process described above enforces constraints in $B$ strictly, *e.g.*, the word "*nice*" will be rejected in an Individuality module looking for the word "*good*", despite their semantic similarity. To broaden the coverage of *strict* matching, we propose *soft* matching for each module, which relaxes the rule body and generates a larger number of matched instances to enhance regularization strength.

Table 2 provides a concrete example. Based on the reference sentence "They prove more distressing than attractive", together with its explanation, a sentence will be matched if and only if it contains distressing and attractive in *strict* matching. In contrast, in *soft* version, distressing can be generalized to depressing and attractive to entertaining.

For Individuality module, we allow synonyms, coreference and morphological/spelling variation of $q_{ref}$ in $x_k$ using a neural model. Given a sentence $x_k = [w_1, w_2, ..., w_m]$, we first encode the sentence with a BERT-Base model [4] and obtain each token's contextualized representations $[v_1, v_2, ..., v_m]$. Given a phrase $q_k = [w_{l1}, ..., w_{l2}]$ in $x_k$, we apply mean pooling over the token representations to get its phrase representation, *i.e.*, $\sum_{j=l_1}^{l_2} v_j/(l_2 - l_1)$. We compute the representation for $q_{ref}$ in an analogous way. The softened individuality module will output a set of candidate spans whose representations have the highest cosine similarity to $q_{ref}$'s.

For Interaction module, we denote the distance between $q_{k,1}$ and $q_{k,2}$ as $d$, and human-specified distance constraint between $q_{ref,1}$ and $q_{ref,2}$ as $d_{ref}$. Instead of strictly following the distance constraint, we compute a score indicating how close $d \leq d_{ref}$ is to being correct: $z = \max(1 - \frac{1}{4}(\frac{d - d_{ref}}{|d_{ref}| + 1})^2, 0)$ if $d > d_{ref}$ and 1 otherwise.

For Compositionality module, soft logic / Lukasiewiczt logic [22] operations are used to aggregate two intermediate scores produced by other modules, *i.e.*, $\text{AND}(z_1, z_2) = \max(z_1 + z_2 - 1, 0); \text{OR}(z_1, z_2) = \min(z_1 + z_2, 1)$.

Using the three atomic modules, we are able to generalize from human-provided explanations to more unlabeled instances in $\mathcal{D}_T$. After the matching process, a matched instance $x_k$ will be associated with a noisy label $y_k$, some refinement advice $r_k$, and a confidence score $z_k$. If strict matching is employed, $z_k$ is set to one. If soft matching is employed, $z_k$ is computed from scores produced by each module. We use $\mathcal{D}_T^{match} = \mathbf{G}(\mathcal{D}_T) = \{(x_k, y_k, r_k, z_k)\}$ to denote this set of instances.

### 3.4 Learning with Explanation-Generalized Data

**Objective for Model Update.** The learning objective for refining $f_S$ is defined as follows.

$$\mathcal{L} = \mathcal{L}' + \alpha(\mathcal{L}^{attr} + \mathcal{L}^{inter}), \tag{1}$$

where $\mathcal{L}'$ is the classification loss using the noisy labels $\{y_k\}$, $\mathcal{L}^{attr}$ and $\mathcal{L}^{inter}$ are regularization terms computed using refinement advice $\{r_k\}$, and $\alpha$ is the hyperparameter to control the strength of the regularization terms. We discuss the selection of $\alpha$ in Sec. 4.4.

With strict matching, the noisy labels $\mathcal{C}_{strict} = \{y_k\}$ and refinement advice $\mathcal{R}_{strict} = \{r_k\}$ are less noisy; with soft matching, the noisy labels $\mathcal{C}_{soft}$ and refinement advice $\mathcal{R}_{soft}$ can cover more training data. As noisy labels and refinement advice incorporate different information, it is preferable to decouple the matching process for the two components, such as using $\mathcal{R}_{soft}$ with $\mathcal{C}_{strict}$.

**Regularization with Refinement Advice.** We denote the attribution and interaction scores produced by the source model before adjustment as $\phi$ and $\varphi$. The refinement advice $r_k$ contains a set of *target* scores $t_p^c$ or $\tau_{p,q}^c$ that suggest how the attribution or interaction scores of specific phrases should be adjusted. Target scores $t_p^c$ and $\tau_{p,q}^c$ of phrases $p, q$ regarding class $c$ are set to 0 if it was suggested to decrease the score, and 1 if suggested to increase it. The regularization terms are the squared $L_2$ distance between current and target importance and interaction scores, summed over all $C$ classes and phrases $p \in x_k$.

$$\mathcal{L}^{attr} = \sum_c^C \sum_{p \in x_k} (\phi^c(p; x_k) - t_p^c)^2; \qquad \mathcal{L}^{inter} = \sum_c^C \sum_{\{p,q\} \in x_k} (\varphi^c(p, q; x_k) - \tau_{p,q}^c)^2. \tag{2}$$

We consider two feature attribution methods for regularization - Integrated Gradient [42] and Sampling and Occlusion [19]. Next, we briefly introduce their formulations and our approach to quantify feature interactions.

**Importance Attribution Score Computation**. Integrated Gradients (IG) computes an importance score of a feature (word) $w_i$ as the integrated gradient along the straight line path from an input sentence $x$ and a neutral baseline $x' = [w_1', w_2', ..., w_m']$ (*e.g.*, a sequence of all padding tokens). Formally, the importance score of a word $w_i$ for a label $c$ is written as, $\phi^c(w_i; x) = (w_i - w_i') \cdot \int_{\alpha=0}^1 \frac{\partial f^c(x' + \alpha \cdot (x - x'))}{\partial w_i}$, where $f^c(\cdot)$ is the model prediction score for the class $c$.

Sampling and occlusion (SOC) assigns importance score of a phrase $p = [w_i, ..., w_j]$ (one or a sequence of words) as the expectation of the prediction change under context replacement within a fix-sized neighboring region $\delta$. We use $f^c(x_{-\delta}; \hat{x}_\delta)$ to denote model prediction when the words in $x$ within $\delta$ region are replaced with sampled $\hat{x}_\delta$; and use $f^c(x_{-\{\delta,p\}}; \hat{x}_\delta; \mathbf{0}_p)$ to further denote the model prediction when the phrase $p$ is replaced with padding tokens $\mathbf{0}_p$. The importance of the phrase $p$ in the input $x$ is computed as, $\phi^c(p; x) = \frac{1}{|\mathcal{S}|} \sum_{\hat{x}_\delta \in \mathcal{S}} [f^c(x_{-\delta}; \hat{x}_\delta) - f^c(x_{-\{\delta,p\}}; \hat{x}_\delta; \mathbf{0}_p)]$, where $\hat{x}_\delta$ are replaced neighboring words sampled from a replacement set $\mathcal{S}$ (*e.g.*, sampled from a language model pre-trained on the training set). Specifically, with the neighboring range $\delta$ set as 0, $\phi^c(p; x) = f^c(x) - f^c(x_{-p}; \mathbf{0}_p)$, the explanation is the same as input occlusion (or leave-one-out) algorithm [54], which is the prediction difference between erasing and keeping $p$ in the sentence.

**Quantifying Feature Interactions.** We borrow the definition of interaction from the cooperative game theory [10]. Under this definition, interaction describes how importance of a phrase changes when other words or phrases are absent or present. Based on the definition, we define the interaction score between two phrases $p$ and $q$ for predicting a class $c$ as

$$\varphi^c(p, q; x) = \phi^c(p; x) - \phi_{-q}^c(p; x), \tag{3}$$

where $\phi_{-q}^c(p, x)$ denotes the importance score of $p$ after masking the phrase $q$ from the sentence.

## 4 Experiments

### 4.1 Experiment Setup

**Datasets.** For hate speech detection, we use Stormfront [2] and HatEval [1] as upstream datasets, and the Gab Hate Corpus (GHC) [20] as the downstream dataset. Stormfront is a corpus collected from the white supremacist website and HatEval is the official Hate Speech dataset from SemEval-2019. Our two upstream datasets contain shorter and simpler sentences than those of GHC, which was

Table 3: **Statistics for explanation solicitation and generalization.** For each dataset pair, we report the annotation information (total time, numbers of labels and explanations written in corresponding time, and explanation yield rate). We also provide the number and precision of matched instances in strict and soft version, and the size of negative sampling instances in hate speech detection. Finally, we include size of $\mathcal{C}_{sample}$, the sets of instances with ground truth labels, which we sample from target domain based on same time cost and use in baseline methods.

| Dataset Pair | Total Time | |labeled| | $|\mathcal{E}|$ | Exp. Yield Rate | |strict| | Prec. of strict | |soft| | Prec. of soft | |balanced| | |sample| |
|---|---|---|---|---|---|---|---|---|---|---|
| HatEval → GHC | 80 mins | 212 | 34 | 16% | 329 | 0.751 | 370 | 0.692 | 1400 | 394 |
| Stormfront → GHC | 94 mins | 285 | 40 | 14% | 237 | 0.717 | 278 | 0.658 | 1600 | 464 |
| AmazonMusic → SST-2 | 15 mins | 47 | 29 | 62% | 1308 | 0.942 | 1737 | 0.917 | 0 | 204 |

collected from a social network with a high rate of hate speech. For sentiment analysis, we first train on AmazonMusic [16], and apply the model to the Stanford Sentiment Treebank-2 (SST-2) dataset [40]. Details of the datasets are described in Appendix A.

**Compared Methods.** We consider two variants of REMOTE in the experiments: (i) using only $\mathcal{R}_{soft}$, and (ii) both $\mathcal{R}_{soft}$ and $\mathcal{C}_{strict}$. We include more experiments REMOTE using both $\mathcal{R}_{soft}$ and $\mathcal{C}_{soft}$ in Appendix D. Detailed experiment settings about hyper-parameters are included in Appendix A.

We compare our method with the following two lines of works. **(1)** *Model transfer* methods: $L^2$ **regularization** [28] places a penalty on the difference between the source and the new model parameters. **Distillation** method [37] adds a loss between prediction of source model and new model to encourage behavior reservation. **(2)** *Explanation-based learning* methods, including **BabbleLabble** [15] and **NExT** [47], use human explanations to generate pseudo-labeled data for model updating. Since our explanation design is different from [15, 47], their matching process is not directly applicable. We adopt our *strict* and *soft* matching method to generate pseudo-labeled data and conduct an ablation study (Sec. 4.4) to compare with: fine-tune with strict-matched data $\mathcal{C}_{strict}$ (as BabbleLabble), and fine-tune with soft-matched data $\mathcal{C}_{soft}$ (as NExT). More experiments compared with other unsupervised model transfer methods [29] are included in Appendix D.

All methods other than REMOTE are trained with *only* labeled instances from $\mathcal{D}_T$. To form fair comparison with the baselines (when annotation time is controlled), we collect a set of labeled instances (denoted as $\mathcal{C}_{sample}$) in the following way: we first estimate the amount of instances that cost the same annotation time as the time used in collecting the explanations (detailed time cost is discussed in Sec. 4.2); then we randomly sample from $\mathcal{D}_T$ this amount of examples along with their ground-truth labels. In addition, we report performance of the source model fine-tuned over the fully labeled $D_T$ set (denoted as $\mathcal{C}_{all}$), referred as **fine-tune** ($\mathcal{C}_{all}$). As an "oracle method", it provides a reference on how much space is left for improvement for source model.

**Evaluation Metrics.** In addition to F1-scores on source domain and target domain test sets, we also evaluate the hate-speech classifiers on the Identity Phrase Templates Test Set [5] (77,000 samples, with 50% of them as "toxic"). We evaluate the unintended biases for identity phrases $T$ by computing the False Positive Rate Difference (FPRD) [5] as $\sum_{t \in T} |FPR - FPR_t|$, where $FPR_t$ is the false positive rate computed upon the subset of samples that contain the term $t$. A smaller value in FPRD suggests that the model is less biased.

### 4.2 Explanation Collection and Generalization

**Explanation Solicitation and Time Cost.** In Sec. 3.2 we introduce three steps in explanation solicitation - providing a label, inspecting the heat-map, writing the explanation. We use SOC [19], a post-hoc explanation algorithm to generate heat-maps for annotators, and estimate the time cost for each step during annotation via our self-designed explanation solicitation interface (Sec. F). It takes on average 12/5/25 seconds for the three steps respectively to collect explanations for hate speech detection, and 4/2/14 seconds for sentiment analysis. We present annotation information in details on three data pairs in Table 3. In hate speech detection, due to difference in toxicity ratios between upstream and downstream datasets, we only collect explanations with noisy label as hateful. We defer the discussion on this choice to the case study on hate/nonhate-labeled rules in Sec. 4.4. To balance the label distribution, we randomly sample some instances in target domain as *non-hate* examples $\mathcal{C}_{balanced}$, the sizes of which are also included in Table 3. Experiment shows that model performance is not sensitive to the different numbers and sampling strategies for negative samples (see Appendix D). Our explanations are annotated by three CS graduate students with hourly wage as $20. To obtain high-quality explanations, two annotators need to verify the explanations written by the other annotator. Full agreement among annotators happened 90+% of time.

Table 4: **Main Results on three pairs of datasets with different language models.** We report F1 scores on source domain and target domain with standard deviation, and FPRD on IPTTS as fairness metric for hate speech task. For each setting, we refine the source model with best F1, and run experiment in three random seeds controlling the order of the data during fine-tuning. The annotation time cost of each dataset pair is provided. Best results are **bold**.

| Dataset | HatEval → GHC (80 mins) | | | Stormfront → GHC (94 mins) | | | AmazonMusic → SST-2 (15 mins) | |
|---|---|---|---|---|---|---|---|---|
| Metrics | Source F1 (↑) | Target F1 (↑) | FPRD (↓) | Source F1(↑) | Target F1 (↑) | FPRD (↓) | Source F1 (↑) | Target F1 (↑) |
| BERT-Large | | | | | | | | |
| Source model | **63.7±0.5** | 31.4±1.4 | 124.3 | **59.5±1.1** | 41.9±1.4 | 17.1 | **92.9±0.2** | 87.7±1.0 |
| Fine-tune ($\mathcal{C}_{sample}$) | 60.8±2.5 | 40.7±0.6 | 175.1 | 49.6±1.8 | 45.0±3.1 | 24.3 | 91.4±1.6 | 88.5±1.1 |
| $L^2$-reg. ($\mathcal{C}_{sample}$) | 58.1±2.9 | 41.8±1.8 | 102.2 | 49.9±1.8 | 45.3±1.9 | 12.2 | 90.5±1.1 | 88.9±1.6 |
| Distillation ($\mathcal{C}_{sample}$) | 63.1±2.5 | 43.3±2.0 | 132.7 | 46.5±2.3 | 48.8±1.1 | 23.4 | 91.9±0.3 | 89.4±0.4 |
| REMOTE ($\mathcal{R}_{soft}$) | 61.5±0.2 | 37.5±0.7 | 55.5 | 56.4±2.0 | 45.7±0.9 | **0.5** | 92.7±0.1 | 89.4±0.2 |
| REMOTE ($\mathcal{R}_{soft} + \mathcal{C}_{strict}$) | 62.0±0.4 | **46.1±1.0** | 15.3 | 49.0±3.4 | **52.2±0.4** | 10.0 | 92.7±0.2 | **90.3±0.2** |
| Fine-tune ($\mathcal{C}_{all}$) | 51.3±5.6 | 52.5±0.4 | 98.0 | 46.0±3.8 | 53.8±1.6 | 142.3 | 92.5±0.2 | 94.4±0.4 |
| RoBERTa-Base | | | | | | | | |
| Source model | **62.7±0.9** | 30.9±1.9 | 61.6 | 57.4±1.2 | 39.6±1.2 | 43.8 | **92.4±0.4** | 87.5±0.9 |
| Fine-tune ($\mathcal{C}_{sample}$) | 61.3±3.0 | 40.8±1.2 | 185.2 | 56.7±2.6 | 40.3±1.9 | **20.7** | 91.0±0.9 | 89.0±0.6 |
| $L^2$-reg. ($\mathcal{C}_{sample}$) | 62.4±2.0 | 42.2±0.8 | 292.7 | 48.8±1.5 | 41.7±0.5 | 46.2 | 90.9±1.0 | 89.0±0.6 |
| Distillation $\mathcal{C}_{sample}$) | 61.8±1.4 | 42.1±1.2 | 152.5 | 48.4±1.6 | 40.9±0.3 | 28.1 | 91.3±0.5 | 89.1±0.5 |
| REMOTE ($\mathcal{R}_{soft}$) | 61.1±0.6 | 40.5±1.1 | **15.9** | 57.0±1.0 | 40.9±0.6 | 36.7 | 92.0±0.2 | 88.5±0.7 |
| REMOTE ($\mathcal{R}_{soft} + \mathcal{C}_{strict}$) | 57.5±0.9 | **44.7±1.0** | 97.8 | **57.6±1.9** | **50.1±1.7** | 77.5 | 91.4±0.2 | **89.5±0.5** |
| Fine-tune ($\mathcal{C}_{all}$) | 51.4±3.2 | 50.6±0.4 | 263.2 | 52.2±4.9 | 50.5±1.5 | 294.0 | 91.2±0.0 | 95.1±0.4 |

**Quality Check of Explanation Generalization.** We generalize the collected explanations on unlabeled instances in target domain in *strict* and *soft* version. To best utilize the collected explanations, we tune the threshold of confidence score in soft matching process from 0.5 to 0.9 based on the performance on dev sets, and set them as 0.7, 0.55 and 0.6 for Stormfront → GHC, HatEval → GHC and Amazon → SST-2 respectively. To prove the effectiveness of our matching neural model, we evaluate the matching quality by the precision of noisy labels. The statistics are shown in Table 3.

## 4.3 Performance Comparison

In comparisons below, we keep the total human annotation time per model fixed, by controlling the number of explanations in REMOTE, and the number of labeled instances in model transfer methods.

**Comparison with Model Transfer Baselines.** Table 4 shows results on three source → target dataset pairs with source models trained using BERT-Large [4] and RoBERTa-Base [31]. REMOTE with $\mathcal{R}_{soft} + \mathcal{C}_{strict}$ outperforms all other methods that have access to $\mathcal{C}_{sample}$ with the same annotation time on target performance. This demonstrates that REMOTE is highly label-efficient compared to the model transfer methods.

**Comparison between REMOTE variants.** REMOTE with just $\mathcal{R}_{soft}$ shows notable target F1 improvement over the source models among all dataset pairs and language models. On top of that, REMOTE with $\mathcal{R}_{soft} + \mathcal{C}_{strict}$ performs consistently better than with just $\mathcal{R}_{soft}$. We can conclude that our generated noisy labels are sufficiently accurate and important in a low-resource situation.

**Fairness and Source Domain Performance.** REMOTE preserves source F1 better than fine-tune ($\mathcal{C}_{all}$) in most cases. We also observe that REMOTE mitigates unintended bias (as FPRD values are reduced) and simultaneously achieves target F1 close to the performance of fine-tune ($\mathcal{C}_{all}$). We attribute this to our design of REMOTE which allows annotators to pinpoint spurious patterns *precisely* with compositional explanations, and our feature-level regularization method which refines the model by teaching *why* an label is correct and reduces unintended biases.

Table 5: **Significant Test on Main Results.** We report p-value between target F1 of REMOTE and each baseline method on every dataset pair in Table 4. The difference is regarded as statically significant when $p \leq 0.05$.

| Dataset | HatEval → GHC | | Stormfront → GHC | | Amazon → SST-2 | |
|---|---|---|---|---|---|---|
| Metrics | P-Value | Sig. or not | P-Value | Sig. or not | P-Value | Sig. or not |
| BERT-Large | | | | | | |
| Fine-tune ($\mathcal{C}_{sample}$) | 0.0013 | yes | 0.0163 | yes | 0.0494 | yes |
| $L^2$-reg ($\mathcal{C}_{sample}$) | 0.0224 | yes | 0.0003 | yes | 0.2071 | no |
| Distillation ($\mathcal{C}_{sample}$) | 0.0959 | no | 0.0073 | yes | 0.0252 | yes |
| RoBERTa-Base | | | | | | |
| Fine-tune ($\mathcal{C}_{sample}$) | 0.0124 | yes | 0.0026 | yes | 0.3297 | no |
| $L^2$-reg ($\mathcal{C}_{sample}$) | 0.0278 | yes | 0.0012 | yes | 0.3297 | no |
| Distillation ($\mathcal{C}_{sample}$) | 0.0449 | yes | 0.0008 | yes | 0.3827 | no |

**Significance Test.** To show the statistical significance between our method and baseline methods, we do an unpaired t-test between REMOTE and each baseline methods. We report the P-values in Table 5. We found that the improvements brought by REMOTE is statistically significant in most cases.

Table 6: **Effectiveness of model regularization technique on BERT-Base.** We compare REMOTE ($\mathcal{R}_{soft}$ + $\mathcal{C}_{strict}$) (based on IG and SOC) with other methods using $\mathcal{C}_{strict}$ to show the effect of $\mathcal{R}_{soft}$.

| Dataset | HatEval → GHC | | | Stormfront → GHC | | | AmazonMusic → SST-2 | |
|---|---|---|---|---|---|---|---|---|
| Metrics | Source F1 (↑) | Target F1 (↑) | FPRD (↓) | Source F1(↑) | Target F1 (↑) | FPRD (↓) | Source F1 (↑) | Target F1 (↑) |
| Source model | 64.2±0.3 | 29.5±2.5 | 115.6 | **57.2±0.7** | 42.1±1.5 | 16.0 | 91.4±0.4 | 83.5±2.5 |
| Fine-tune ($\mathcal{C}_{strict}$) [15] | 60.3±1.4 | 45.1±2.2 | 80.2 | 42.0±1.6 | 49.0±0.5 | 59.2 | 90.7±0.1 | 86.0±0.6 |
| $L^2$-reg ($\mathcal{C}_{strict}$) | 62.7±1.1 | 46.3±0.2 | 77.1 | 46.5±0.7 | 49.9±0.8 | 86.4 | 90.7±0.3 | 86.8±0.6 |
| Distillation ($\mathcal{C}_{strict}$) | 63.2±1.0 | 46.3±0.9 | 65.4 | 46.4±1.3 | 49.4±1.1 | 49.7 | 90.6±0.5 | 86.7±1.0 |
| Fine-tune ($\mathcal{C}_{soft}$) [47] | 62.4±1.8 | 45.4±2.0 | 57.9 | 49.6±4.1 | 47.9±0.6 | 164.0 | 90.4±0.9 | 86.3±0.4 |
| REMOTE ($\mathcal{R}_{soft}$ + $\mathcal{C}_{strict}$) w. IG | **64.2±0.4** | **47.2±1.3** | 129.5 | 51.4±4.6 | 49.5±1.1 | 12.8 | **91.2±0.1** | 87.0±0.7 |
| REMOTE ($\mathcal{R}_{soft}$ + $\mathcal{C}_{strict}$) w. SOC | 63.2±0.6 | 46.6±1.1 | **49.0** | 49.4±1.9 | **51.1±1.6** | 34.6 | 91.1±0.2 | **87.3±0.1** |
| Fine-tune ($\mathcal{C}_{all}$) | 60.0±2.3 | 51.5±0.9 | 333.3 | 46.9±2.4 | 52.9±1.0 | 115.0 | 90.4±0.1 | 92.9±0.4 |

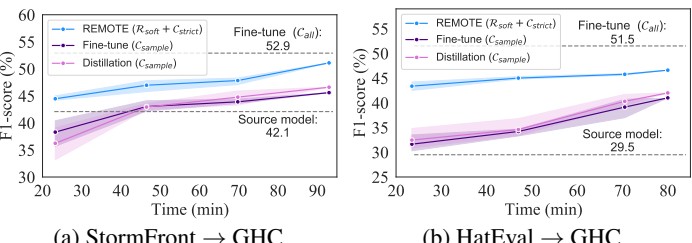

Figure 3: **Label Efficiency Study on BERT-Base.**

(a) StormFront → GHC

(b) HatEval → GHC

Figure 4: Performance on HatEval → GHC on BERT-Base using different number of explanations.

**Across Different Language Models.** We conduct experiments across different language models to study whether improvement brought by REMOTE is consistent. In Table 4, performance for both BERT-Large and RoBERTa-Base shows that incorporating noisy labels and regularization advice can generally help, and it can be widely applied to different pre-trained language models. We also include additional experiments using BERT-Base [4] and BiLSTM+Attention in Appendix C.

**Study on Label Efficiency.** We further analyze the performance trends of different methods when varying the annotation time. Figure 3 shows that with the same amount of annotation time, REMOTE consistently outperforms the fine-tuning and distillation baseline, demonstrating a better label efficiency. Within 30 minutes of annotation, REMOTE improves the F1 score on the downstream dataset significantly for HateEval → GHC. This indicates that even a small number of compositional explanations can provide strong supervision towards adapting the model to the target domain.

## 4.4 Performance Analysis

**Effectiveness of Explanation Regularization.** In Table 6, REMOTE ($\mathcal{R}_{soft}$ + $\mathcal{C}_{strict}$) has lower FPRD and higher target F1 scores compared with fine-tuning or performing model transfer using noisy labels ($\mathcal{C}_{strict}$ or $\mathcal{C}_{soft}$). We observe that $\mathcal{R}_{soft}$ plays an essential role in reducing unintended biases. As $\mathcal{R}_{soft}$ contains specified information regarding identity groups, it enables the model to better capture the context involving identity phrases, and thus improves the fairness of the model. We include comparison between solely using $\mathcal{R}_{strict}$ and $\mathcal{R}_{soft}$ in Appendix D. $\mathcal{R}_{soft}$ always leads to better performance than $\mathcal{R}_{strict}$. With soft-matching, regularization advice is applied to more instances and thus take more effect.

**Effectiveness of Noisy Labels.** To show the effectiveness of the noisy labels generated by REMOTE, we further compare the performance among fine-tuning with $\mathcal{C}_{sample}$ (labeled instances randomly selected from $\mathcal{D}_T$), $\mathcal{C}_{strict}$ and $\mathcal{C}_{soft}$ (noisy labels generated in different matching versions) on hate speech detection. It shows that models fine-tuned with noisy labels can yield better performance than ground-truth labels. It is partly because that $\mathcal{C}_{sample}$ is ran-

Table 7: **Comparison on fine-tuning with different datasets.** We compare performance of simply fine-tuning the source model on BERT-Base with $\mathcal{C}_{sample}$, $\mathcal{C}_{strict}$ and $\mathcal{C}_{soft}$ on hate speech detection.

| Dataset | Stormfront→GHC | | HatEval→GHC | |
|---|---|---|---|---|
| Metrics | F1 (↑) | FPRD (↓) | F1 (↑) | FPRD (↓) |
| Source model | 42.1±1.5 | 16.0 | 29.5±2.5 | 115.6 |
| Fine-tune ($\mathcal{C}_{sample}$) | 41.0±0.1 | 302.6 | 45.6±0.1 | 20.3 |
| Fine-tune ($\mathcal{C}_{strict}$) | 45.1±2.2 | 80.2 | **49.0±0.5** | 59.2 |
| Fine-tune ($\mathcal{C}_{soft}$) | **45.4±2.0** | 57.9 | 47.9±0.6 | 164.0 |

domly sampled and does not target cases with spurious patterns. In contrast, $\mathcal{C}_{strict}$ and $\mathcal{C}_{soft}$ generalize collected explanations to unlabeled instances to create hateful instances, and then randomly samples some unlabeled instances from $\mathcal{D}_T$ as non-hateful instances. Regarding quantity, the numbers of matched instances are smaller than the size of $\mathcal{C}_{sample}$, but the sizes of $\mathcal{C}_{strict}$ or $\mathcal{C}_{soft}$ (generalized hateful instances and sampled non-hateful instances) are larger than $\mathcal{C}_{sample}$. Overall, quality and quantity of data both contribute to the final performance.

**Performance Change from Increasing the Number of Explanations.** We investigate the performance in HatEval → GHC setting using different number of explanations. We present the results in Fig. 4. Note that we randomly sample the subset of explanations for three times to reduce variance. We observe that that the performance continuously grows when more explanations are introduced. This indicates that model can learn more diverse knowledge from more explanations. We also conduct the same set of experiments for Stormfront → GHC setting. Results are included in Appendix D.

**Ablation Study on Attribution and Interaction.** To demonstrate the necessity of regularizing both attribution and interaction, we conduct ablation experiments that regularize them separately. Results in Table 8(b) show that both attribution and interaction can contribute to performance improvements and bias mitigation. Regularization with both attribution and interaction achieves higher F1 score in target domain than regularizing with only one of them.

Table 8: **Ablation study on hate speech detection on BERT-Base with REMOTE($\mathcal{R}_{soft}$).**

(a) Ablation study on hate/nonhate-labeled rules.

| Dataset | Stormfront→GHC | | Hateval→GHC | |
|---|---|---|---|---|
| Metrics | F1 (↑) | FPRD (↓) | F1 (↑) | FPRD (↓) |
| Source Model | 42.1±1.5 | 16.0 | 29.5±2.5 | 115.6 |
| Nonhate-labeled rules | 43.2±0.4 | 9.0 | 32.1±0.5 | 175.3 |
| All rules | 43.2±0.4 | 8.2 | 35.7±2.0 | 173.0 |
| Hate-labeled rules | **45.7±1.4** | 12.6 | **39.9±4.4** | 56.2 |

(b) Case study on attribution (a.) and interaction (i.).

| Dataset | Stormfront→GHC | | HatEval→GHC | |
|---|---|---|---|---|
| Metrics | F1 (↑) | FPRD (↓) | F1 (↑) | FPRD (↓) |
| Source model | 42.1±1.5 | 16.0 | 29.5±2.5 | 115.6 |
| Reg. with a. | 44.6±1.6 | 7.2 | 36.8±4.7 | 39.8 |
| Reg. with i. | 43.2±0.1 | 8.5 | 31.8±0.5 | 170.7 |
| Reg. with a. & i. | **45.7±1.4** | 12.6 | **39.9±4.4** | 56.2 |

**Case Study on Hate-labeled Rules and Nonhate-labeled Rules for GHC.** To reveal the reason of only collecting hate-labeled explanations for GHC, we show the ablation results in Table 8(a). Note that, here "label" is the noisy label given by human annotators. Besides the hate-labeled rules, we also collect nonhate-labeled rules, and conduct experiments with them separately and together. We can see that regularization with only hateful rules has the best performance on the target domain and has low unintended biases.

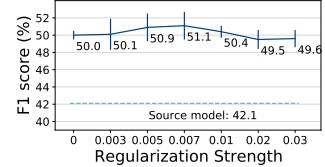

**Sensitivity Study on Regularization Strength.** We tune $\alpha$ between 0.003 and 0.03 in the experiments of REMOTE ($\mathcal{R}_{soft} + \mathcal{C}_{strict}$) and report the results in Fig. 5. REMOTE is not very sensitive to $\alpha$, and improves the source model performance with a wide range of values. Results on another dataset pair are included in Appendix D.

Figure 5: Sensitivity of reg. strength on Stormfront to GHC.

**Discussion on Optimisation Time Cost.** The time cost for computing the loss in Eq. (2) depends on the proportion of phrases that are given target scores (by generalizing the human-provided explanations). In our implementation, only the phrases that are given target scores will be traversed when computing the loss. Typically, very few phrases in a matched sentence would have target scores assigned. For example in the HatEval → GHC setting, 291 out of the 370 soft-matched instances have only one phrase/phrase pair with target score assigned. The main time cost comes from computing the importance/interaction scores in line 264 and Eq 3. With the hardware and training details specified in Appendix A, it usually takes 50s for one iteration with $\mathcal{R}_{soft}$ loss and 30s without it.

## 5 Conclusion

In this work, we introduce a novel framework to refine a trained language model with human-provided compositional explanations. To break the bottlenecks of previous explanation regularization methods, our method (1) supports more diverse ways of regularization (*i.e.*, both feature attributions and interactions) and (2) broadens regularization coverage by generalizing explanations to unlabeled instances. We believe our work opens up a new way to communicate human knowledge to model learning and refinement, and explores the possibility of supervising model features, as opposed to the traditional paradigm of supervising a model with large-scale labeled examples. Extensive experiments demonstrate that our framework can improve source model's performance on different tasks and reduce unintended bias. Future work may expand on our method by adapting it to more challenging tasks such as reading comprehension and language generation, or by incorporating human feedback in multiple turns in an active learning setting.

## Acknowledgments and Disclosure of Funding

This research is supported in part by the Office of the Director of National Intelligence (ODNI), Intelligence Advanced Research Projects Activity (IARPA), via Contract No. 2019-19051600007, the DARPA MCS program under Contract No. N660011924033 with the United States Office Of Naval Research, the Defense Advanced Research Projects Agency with award W911NF-19-20271, and NSF SMA 18-29268. The views and conclusions contained herein are those of the authors and should not be interpreted as necessarily representing the official policies, either expressed or implied, of ODNI, IARPA, or the U.S. Government. We would like to thank all the collaborators in USC INK research lab for their constructive feedback on the work.

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
