# A    Experiment Details for Reproducibility

**Dataset.**    For experiments on hate speech detection, we download the Gab Hate Corpus (GHC) from [20][2]. Stormfront and HatEval datasets are downloaded from [2][3] and [1][4]. For sentiment analysis, AmazonMusic is released by [16][5] and SST-2 dataset can be downloaded from [40][6].

For Gab Hate Corpus (GHC) dataset, we randomly re-split the dataset into train/dev/test sets by the ratio 8:1:1. For all the other datasets, we follow their original train/dev/test splits. Target training set is used as the "target domain data" after removing the ground-truth labels. Target dev set is used for early stopping when updating the target models and also for tuning model hyper-parameters. Target test set is used for evaluating "target F1" metric. We fine-tune a pre-trained language model (e.g., BERT-Base) over the source training set to generate the source model. Source dev set is used for early stopping when training the source model. Source test set is used for evaluating the "source F1" metric for the updated models. Statistics of each dataset pair are included in Table 9.

Table 9: Statistics for the dataset pairs.

| Dataset Pair | Source train/dev/test | Target train/dev/test |
|---|---|---|
| HatEval → GHC | 9000 / 1000 / 3000 | 22132 / 2766 / 2767 |
| Stormfront → GHC | 7896 / 978 / 1998 | 22132 / 2766 / 2767 |
| AmazonMusic → SST-2 | 3000 / 300 / 8302 | 67349 / 872 / 1821 |

**Implementation and Infrastructure.**    All our experiments are implemented with Transformers library [50][7]. All experiments are done with one single GPU. We use NVIDIA Quadro RTX 8000 for large-sized language models (i.e., BERT-Large) and NVIDIA GeForce RTX 2080 Ti for other models (e.g., BERT-Base, RoBERTa-Base, Bi-LSTM+Attention).

**Hyperparameters.**    We use Adam [23] optimizer throughout all the experiments. Batch size is set to be 32 in all experiments for all the methods. We conduct grid search on learning rate and regularization strength for each experiment using the target dev set. For REMOTE ($\mathcal{R}$) and all the baselines, learning rate is selected from the range {5e-6, 8e-6, 1e-5, 2e-5, 3e-5, 4e-5, 5e-5, 6e-5, 7e-5}. Regularization strength (when applicable) is selected from {0.01, 0.02, 0.03, 0.04, 0.05}. For REMOTE ($\mathcal{R} + \mathcal{C}$), learning rate is selected from the range {1e-5, 2e-5, 3e-5, 4e-5}. Regularization strength is selected from {0.003, 0.005, 0.007, 0.01, 0.02, 0.03}.

**Early Stopping.**    We evaluate the model performance over target dev set every 400 steps for REMOTE ($\mathcal{R} + \mathcal{C}$) and every 100 steps for REMOTE ($\mathcal{R}$) and all the baselines. The training will be early-stopped when the target dev F1 stops improving for 10 iterations, and the learning rate is halved when the dev F1 stops improving for 5 iterations.

**Multiple Runs.**    For every experiment setting, we select the best configuration of hyper-parameters based on the target dev set F1 using one random seed. Then we train the model using this hyper-parameter configuration with two additional random seeds and report the mean and standard deviation.

**Label Balancing for Hate Speech Tasks.**    For hate speech tasks, due to the unbalanced ratio of negative and positive examples (approximately 10:1), we re-weight the training loss so that positive examples are weighted 10 times as negative examples for all the models.

# B    Details on Explanation Parsing

**Lexicon Details.**    To help the parser understand the collected natural language explanations, we define a lexicon *str2predicate* that maps 301 raw words/phrases into 83 predicates. In addition,

---

[2]https://osf.io/nqt6h/

[3]https://github.com/Vicomtech/hate-speech-dataset

[4]https://competitions.codalab.org/competitions/19935

[5]https://sites.google.com/a/eng.ucsd.edu/ruining-he/

[6]https://dl.fbaipublicfiles.com/glue/data/SST-2.zip

[7]https://github.com/huggingface/transformers

we define a lexicon *predicate2func* that maps the predicates to 25 functions. For example, when human annotator write the word political, politician, religious or nationality in their explanation, the word will first be mapped to the predicate $NorpNER. NORP is the shorthand for "nationalities or religious or political groups". Then $NorpNER will be processed and combined with other predicates in the same sentence (using CCG Parser), and the final parsing result may map it to the function @NER(NORP), which will identify whether a given word is indeed a NORP named entity. We have included these lexicons in our code.

**Parser Evaluation and Modification.** As a pre-validation of the parsed rule, we first execute it on its reference instance $x_{ref}$ and discard the rule if the execution outcome of the rule head $B$ over $x_{ref}$ is False (i.e., the parsed rule cannot match the reference instance $x_{ref}$ where the explanation was solicited from). This step ensures the quality of explanations and parsed rules. After a first run of parsing of the explanations, we further evaluate the quality of the parsed rules to ensure the parser is working properly. If the parsed rules are not equivalent to their original explanations, we modify the parser's lexicon to adjust the parsing results. For example, if the annotator wrote "X is religion", while "religion" is not in the pre-defined *str2predicate* lexicon, the sentence will be ignored by CCG parser at first. To correct the mistake, we can add "religion" to the lexicon. About 85% of explanations can be parsed correctly in the first time (as we manually inspected). After updating the lexicon, the parsing will reach 100% accuracy on all the collected explanations.

**Parser Evaluation on Out-Of-Domain Data.** To understand how well our parser can generalize to out-of-domain data, we further collected 116 explanations following our annotation guideline, on held-out data from Stormfront $\rightarrow$ GHC and HatEval $\rightarrow$ GHC. Note that this set of instances are disjoint from the annotation sets used in our reported experiments. We present the parsing results to human annotators for verification. This evaluation shows that the accuracy of semantic parser can reach 92.2% on the held-out data (without any update to the parser). Therefore, we believe our semantic parser is reliable when applied to out-of-domain data. We also admit that the parse may encounter unseen vocabulary and typos when human annotators are not strictly following the annotation guideline.

## C   Results on Refining BERT-Base and Bi-LSTM+Attention

Table 10: **Results on three pairs of datasets with BERT-Base and BiLSTM+Attention.** We report F1 scores on source domain and target domain, and FPRD on IPTTS as fairness metric for hate speech task. Best results are **bold**. The annotation time cost of each dataset pair is provided. We use SOC for feature attribution.

| Dataset | HatEval $\rightarrow$ GHC (80 mins) | | | Stormfront $\rightarrow$ GHC (94 mins) | | | AmazonMusic $\rightarrow$ SST-2 (15 mins) | |
|---|---|---|---|---|---|---|---|---|
| Metrics | Source F1 (↑) | Target F1 (↑) | FPRD (↓) | Source F1(↑) | Target F1 (↑) | FPRD (↓) | Source F1 (↑) | Target F1 (↑) |
| BERT-Base | | | | | | | | |
| Source model | 64.2±0.3 | 29.5±2.5 | 115.6 | 57.2±0.7 | 42.1±1.5 | 16.0 | 91.4±0.4 | 83.5±2.5 |
| Fine-tune ($\mathcal{C}_{sample}$) | 58.0±5.1 | 41.0±0.1 | 302.6 | **56.4±0.4** | 45.6±0.1 | 20.3 | 88.9±0.6 | 86.7±1.0 |
| $L^2$-reg ($\mathcal{C}_{sample}$) | 59.8±4.2 | 41.3±0.7 | 278.7 | 55.8±0.9 | 46.8±1.2 | 26.8 | 89.0±0.6 | 86.8±0.6 |
| Distillation ($\mathcal{C}_{sample}$) | 60.8±5.1 | 42.4±1.6 | 315.4 | 54.4±2.0 | 46.6±1.4 | 112.6 | 87.4±0.9 | 86.7±1.0 |
| REMOTE ($\mathcal{R}_s$) | 63.5±0.7 | 39.9±4.4 | 56.2 | 49.9±3.5 | 45.7±1.4 | **12.6** | 91.1±0.0 | 85.3±0.1 |
| REMOTE ($\mathcal{R}_s + \mathcal{C}_h$) | **63.2±0.6** | **46.6±1.1** | 49.0 | 47.4±1.9 | **51.1±1.6** | 34.6 | **91.1±0.2** | **87.3±0.1** |
| Fine-tune ($\mathcal{C}_{all}$) | 60.0±2.3 | 51.5±0.9 | 333.3 | 46.9±2.4 | 52.9±1.0 | 115.0 | 90.4±0.1 | 92.9±0.4 |
| BiLSTM + Attention | | | | | | | | |
| Source model | 60.5±0.7 | 20.2±1.2 | 115.2 | 44.7±2.2 | 26.3±4.2 | 157.2 | 81.1±1.5 | 54.0±5.5 |
| Fine-tune ($\mathcal{C}_{sample}$) | 60.8±0.2 | 23.7±0.8 | 71.9 | 42.9±2.7 | 29.1±2.3 | 201.8 | **78.9±0.4** | 61.0±1.3 |
| $L^2$-reg ($\mathcal{C}_{sample}$) | 60.4±0.3 | 24.2±0.1 | 21.3 | 44.3±0.3 | 30.1±0.9 | 157.3 | 78.7±0.4 | 62.6±1.1 |
| Distillation ($\mathcal{C}_{sample}$) | 60.1±0.5 | 24.3±0.8 | 16.4 | **45.2±0.9** | 29.5±0.9 | 151.3 | 78.6±0.6 | 61.6±1.3 |
| REMOTE ($\mathcal{R}_{soft}$) | **61.1±0.1** | 27.0±0.4 | 69.3 | 36.8±5.1 | 31.5±0.9 | **18.6** | 78.5±0.7 | 64.9±1.5 |
| REMOTE ($\mathcal{R}_{soft} + \mathcal{C}_{strict}$) | 58.7±1.5 | **31.3±2.5** | **7.4** | 42.2±2.2 | **33.5±0.4** | 65.2 | 76.9±0.5 | **66.1±0.6** |
| Fine-tune ($\mathcal{C}_{all}$) | 58.5±1.7 | 38.5±1.7 | 124.2 | 42.5±2.0 | 44.5±0.2 | 323.3 | 79.2±0.2 | 81.7±1.1 |

As additional results for performance comparison, we conduct experiments on BERT-Base and Bi-LSTM+Attention under the same setting as in Table 4 and summarize the results in Table 10. We observe similar patterns as in Table 4. The results show that REMOTE can cope with different language models and obtain consistent improvement over the baselines – it constantly yields the best target performance and fairness among all compared methods, while preserves source performance better than fine-tune ($\mathcal{C}_{all}$) in most cases.

Table 11: **Results with REMOTE ($\mathcal{R}_{soft} + \mathcal{S}_{soft}$).** By using $\mathcal{C}_{soft}$ as noisy labels rather than $\mathcal{C}_{strict}$, the target domain F1 scores are lower in most cases.

| Dataset | HatEval → GHC (80 mins) | | | Stormfront → GHC (94 mins) | | | AmazonMusic → SST-2 (15 mins) | |
|---|---|---|---|---|---|---|---|---|
| Metrics | Source F1 (↑) | Target F1 (↑) | FPRD (↓) | Source F1(↑) | Target F1 (↑) | FPRD (↓) | Source F1 (↑) | Target F1 (↑) |
| BERT-Large | | | | | | | | |
| Source model | **63.7±0.5** | 31.4±1.4 | 124.3 | **59.5±1.1** | 41.9±1.4 | 17.1 | **92.9±0.2** | 87.7±1.0 |
| REMOTE ($\mathcal{R}_{soft} + \mathcal{C}_{strict}$) | 62.0±0.4 | 46.1±1.0 | 15.3 | 49.0±3.4 | **52.2±0.4** | **10.0** | 92.7±0.2 | **90.3±0.2** |
| REMOTE ($\mathcal{R}_{soft} + \mathcal{C}_{soft}$) | 61.8±0.2 | **47.4±0.9** | **11.2** | 50.7±1.9 | 51.6±2.7 | 17.8 | 92.7±0.0 | 89.5±0.1 |
| Fine-tune ($\mathcal{C}_{all}$) | 51.3±5.6 | 52.5±0.4 | 98.0 | 46.0±3.8 | 53.8±1.6 | 142.3 | 92.5±0.2 | 94.4±0.4 |
| RoBERTa-Base | | | | | | | | |
| Source model | **62.7±0.9** | 30.9±1.9 | 61.6 | 57.4±1.2 | 39.6±1.2 | 43.8 | **92.4±0.4** | 87.5±0.9 |
| REMOTE ($\mathcal{R}_{soft} + \mathcal{C}_{strict}$) | 57.5±0.9 | **44.7±1.0** | 97.8 | **57.6±1.9** | **50.1±1.7** | 77.5 | 91.4±0.2 | **89.5±0.5** |
| REMOTE ($\mathcal{R}_{soft} + \mathcal{C}_{soft}$) | 59.2±1.4 | 44.0±1.0 | **8.6** | 46.0±2.6 | 49.6±0.8 | **19.2** | 91.2±0.3 | 89.1±0.9 |
| Fine-tune ($\mathcal{C}_{all}$) | 51.4±3.2 | 50.6±0.4 | 263.2 | 52.2±4.9 | 50.5±1.5 | 294.0 | 91.2±0.0 | 95.1±0.4 |

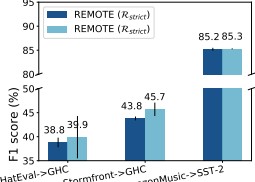 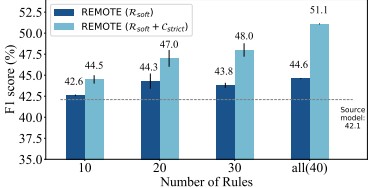 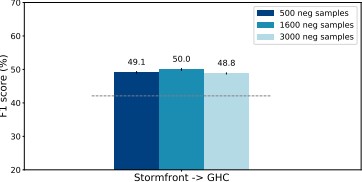

Figure 6: REMOTE with $\mathcal{R}_{strict}$ and $\mathcal{R}_{soft}$ on BERT-Base.

Figure 7: Stormfront → GHC performance with different numbers of explanations.

Figure 8: Sensitivity study on different negative samplings on Stormfront → GHC.

## D    Additional Experiments for Performance Analysis

**Ablation Study on Refinement with Strict/Soft-Matched Labels.** Table 11 compares the model performance of REMOTE using $\mathcal{R}_{soft} + \mathcal{C}_{strict}$ and $\mathcal{R}_{soft} + \mathcal{C}_{soft}$. It shows that refinement with strict-matched labels ($\mathcal{C}_{strict}$) often outperform the model refined with soft-matched labels ($\mathcal{C}_{soft}$). It shows that model refinement is sensitive to the precision of noisy labels so we decide to use strict-matched labels in our main experiments.

**Ablation Study on Regularization with Strict/Soft-Matched Instances.** Figure 6 compares the target F1 performance of REMOTE using $\mathcal{R}_{strict}$ and $\mathcal{R}_{soft}$ as regularization advice without noisy labels. For the three dataset pairs, $\mathcal{R}_{soft}$ always yields better performance than $\mathcal{R}_{strict}$. With soft-matching, regularization advice are generalized to more instances and thus take more effect. Therefore, we present results on REMOTE the regularization term from $\mathcal{R}_{soft}$ in the main experiments.

**Performance changes by Varying the Number of Explanations.** In addition to the setting reported in Fig. 4, we also conduct experiment on Stormfront → GHC by varying the number of explanations. In Fig. 7, result shows that model performance continuously grows when more explanations are introduced, which is the same pattern as in Fig. 4.

**Sensitivity Study on Negative Samples.** For hate speech detection experiments, we randomly sample (unlabeled) examples from the training set and treat them as "negative" (or "non-hate") examples to balance the label ratio. We control the number of sampled instances in the Stormfront → GHC setting (as 500, 1600, and 3000 instances), and show the results in Fig. 8. For each number, we randomly sample the instances for 3 times. We observed that performance is not very sensitive to the number of negative samples included. Our main results are based on sampling 1,600 negative instances, which has slightly better performance among all.

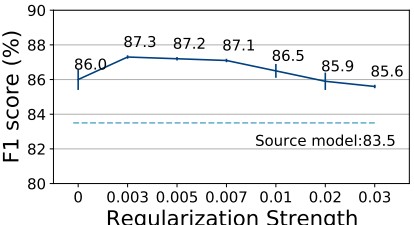

Figure 9: Sensitivity of reg. strength on Amazon to SST-2.

**Sensitivity Study on Reg. Strength.** We report results on sensitivity of model refinement to regularization strength on dataset pair Amazon → SST-2 in Figure 9. We tune reg. strength from 0.0003 to 0.03 in the experiments of REMOTE ($\mathcal{R}_{soft} + \mathcal{C}_{strict}$). We conclude that REMOTE is not

sensitive to the selection of regularization strength, and the refined model constantly perform better than source model.

**Comparison with Unsupervised Model Adaptation Method.** Recent works have studied how to adapt a model trained with source dataset to a target domain only with unlabeled target instances and no source data. This setting is known as *unsupervised model adaptation* (UMA). We apply a popular UMA method SHOT [29] on our task and report the results in

Table 12: **Comparison with unsupervised model adaptation method.** We compare REMOTE ($\mathcal{R}_{soft} + \mathcal{C}_{strict}$) (based on IG and SOC) with a representative UMA method SHOT[29].

| Dataset | HatEval $\to$ GHC | Stormfront $\to$ GHC | Amazon $\to$ SST-2 |
|---|---|---|---|
| Metrics | Target F1 ($\uparrow$) | Target F1 ($\uparrow$) | Target F1 ($\uparrow$) |
| Source model | 29.5±2.5 | 42.1±1.5 | 83.5±2.5 |
| SHOT [29] | 25.4±1.6 | 28.8±1.9 | 61.0±16.1 |
| REMOTE w. IG | **47.2±1.3** | 49.5±1.1 | 87.0±0.7 |
| REMOTE w. SOC | 46.6±1.1 | **51.1±1.6** | **87.3±0.1** |
| fine-tune ($\mathcal{C}_{all}$) | 51.5±0.9 | 52.9±1.0 | 92.9±0.4 |

Table 12. We found that the SHOT harms rather than improves model performance in the target domain. SHOT was originally proposed for computer vision tasks such as object detection and digit recognition. We conjecture that SHOT, an approach proposed for computer vision tasks, may not be directly applicable to a different modality (*i.e.*, natural language). We defer thorough study on extending UMA method to our problem as future work.

**Ablation Study on Soft Version of Interaction Module.** To understand the effect of "softening" change to the Interaction module , we conduct an ablation study on Stormfront $\to$ GHC using BERT-base, as shown in table 13. Specifically, we set all other modules in REMOTE as their soft versions but only the Interaction module as its strict version, and compare it with "REMOTE (all soft)" to show

Table 13: **Ablation Study on soft version of INTER module.** For StormFront $\to$ GHC, we report results on REMOTEn soft version and replace the softened INTER module with strict version on BERT-Base.

| Dataset | Stormfront$\to$GHC | | |
|---|---|---|---|
| Metrics | Source F1 ($\uparrow$) | Target F1 ($\uparrow$) | FPRD ($\downarrow$) |
| Source model | **57.2±0.7** | 42.1±1.5 | 16.0 |
| REMOTEall soft but INTER) | 51.0±0.9 | 44.8±0.4 | **5.2** |
| REMOTEall soft) | 49.9±3.5 | **45.7±1.4** | 12.6 |

the effectiveness of softening the Interaction. Results show that softening the interaction module is an important operation in generalizing explanations to a broader set of unlabeled instances (as discussed in Sec 3.3). When we replace the softened version of Interaction with its strict counterpart, the performance significantly drops.

# E    Details about Individuality modules

In this section we introduce details about how to conduct strict matching via Individuality module. Given the reference sentence $x_{ref}$ and a word $q_{ref}$ in it, the module finds a word $q_k$ in the unlabeled instance $x_k$, where $q_k$ has the same semantic type or plays the same grammatical role with $q_{ref}$. The model determines whether $q_{ref}$ and $q_k$ have the same semantic type according to their named entity types, their sentiment types, and whether they are both identity phrases or hateful words. For sentiment labels, we use the subjectivity lexicon [49] to decide if a word is positive, negative or neutral. For identity phrases, we take the list in [19] which contains group identifiers such as "*women*" or "*black*". In addition, because we aim at hate speech task, we use a list of hateful words obtained from HateBase[8]. We've uploaded the aformentioned lists together with our code, except for hateful word list due to license issue. As for the grammatical roles of $q_{ref}$ and $q_k$, the model compares their relations to the dependency trees and constituency trees of the sentences they are from (dependency parser and constituency parser implemented in *spaCy*[9]).

# F    Explanation Solicitation Interface

The screenshots of the interface for explanation solicitation of the hate speech detection task are included in Fig. 10. The annotators are given the following instructions:

**1) Read the sentence and provide a label.** You will see a sentence on the top of each page. Please decide the label of the sentence, and fill in a number (0 for non-hateful, and 1 for hateful) in the blank.

---

[8]https://hatebase.org/search_results

[9]https://spacy.io/

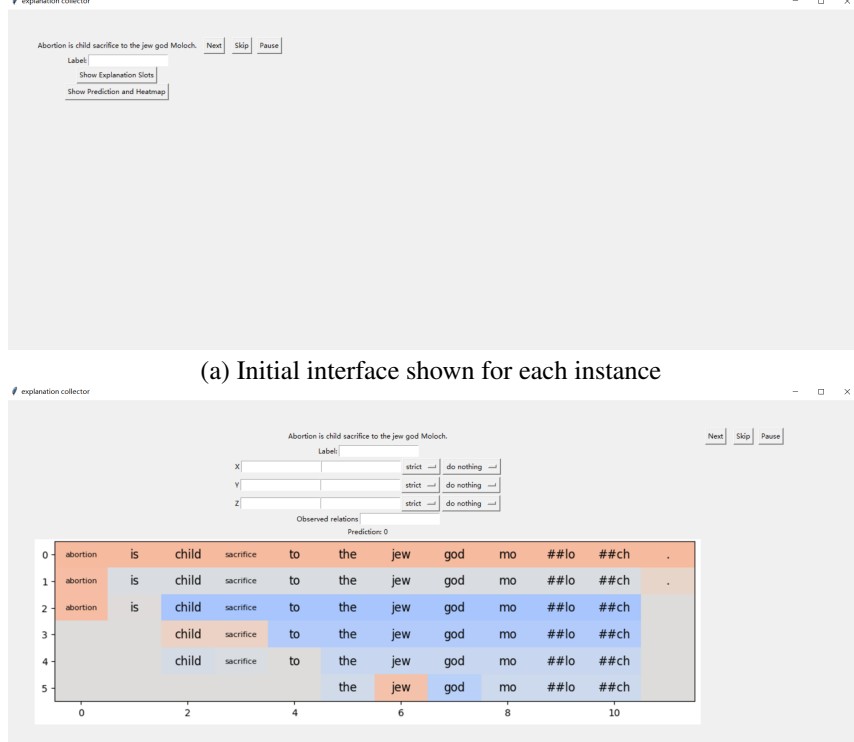

(a) Initial interface shown for each instance

(b) Heat-map and explanation slots shown after annotators select buttons

Figure 10: Interface for Explanation Solicitation

**2) Inspect the heat-map.** Please select the "Show Prediction and Heatmap" button. You will see a predicted label, and a heat-map. In the heat-map, if a word span is considered to be related to hate speech, it is marked as red.

**3) Write an explanation if a spurious pattern is found.** If you think the predicted label and the heat-map do not align with your interpretation of the sentence, please select the "Show Explanation Slots" button. Please specify at least one phrase that you believe the color in the heat-map should be changed, and fill in the phrases in the left-hand side slots, and select the actions that you suggest in the rightmost drop-down lists. You can explain the characteristic of this phrase in its neighboring slot, such as the sentiment or the part-of-speech tag. If you choose to describe the characteristic, please select the "soft" option in the neighboring drop-down list. You can also describe the relations between the specified phrases in the "Observed relations" slot.

**Additional Instructions.** The duration of annotation process is measured. Please press the "Pause" button or close the window when you decide to leave. The program will save your progress. Once you finish filling the slots, you can select the "Next" button to proceed to the next sentence. You can also select "Skip" to skip the current sentence.Though NLP knowledge about post-hoc explanation scores and text classification tasks is required to use our system, we believe the heatmap-based annotation interface is accessible to lay users.

## G  Case Study

Fig. 11 demonstrates an example to show the effect of model refinement. The corresponding human explanation is on the top. The first heat-map is produced by SOC algorithm based on source model $f_S$, and the second heat-map is based on $f_T$, the refined model. We observe that the refined model makes correct prediction. The differences between the two maps demonstrate that attribution scores are adjusted according to human explanations, as expected.

## H    Societal Impact

Our approach can be widely applied to adapt trained text classifiers without accessing upstream data. Therefore, for social media websites aiming to detect hate speech, or customer services doing sentiment analysis based on customer feedback, different service providers can share trained neural models without leaking users' private information. This also reduces the need of collecting a large amount of users' data to fine-tune trained text classifiers.

In addition, as the natural language explanations provided are precise enough, the unintended biases of an existing model can be reduced after refinement. However, if the explanations are not inspected by other annotators or do not pass quality check, but are still applied to refine the model, the model may be at risk of biased explanations that are maliciously injected. To avoid this situation, human annotators need to be required to reach agreement on explanations.

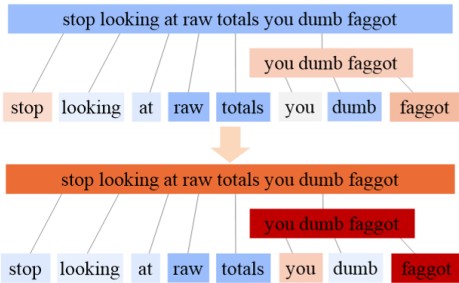

Figure 11: **Post-hoc explanation heat-maps before and after model refinement.** Top: Before regularization; Bottom: After RE-MOTE regularization. Word spans contributing to hate are red, and non-hate ones are blue.