# OpenReview forum: "Refining Language Models with Compositional Explanations"
_NeurIPS.cc/2021/Conference — NeurIPS 2021 Spotlight_

### Official Review · Reviewer_fGUd · 2021-07-15

**Rating:** 7
**Confidence:** 4

**Summary:**

see main review

**Limitations And Societal Impact:**

see main review

**Main Review:**

This paper describes a procedure for improving the fine-tuning of deep network
by eliciting fine-grained natural langauge explanations of model decisions from
human annotators. On a set of unlabeled examples, annotators identify a set of
predicative or relational features that provide evidence for a specific label,
and describe the relationship between these features and labels in language.
Descriptions are then parsed into logical rules, which are applied to a large
corpus of unlabeled data. Finally, the model is fine-tuned on the
(semi-automatically) labeled data with an "attribution loss" that encourages
off-the-shelf feature attribution methods to pick out the labeled features for
the correct label (i.e. for the model to be right for the right reason).
Given a fixed annotation budget, this method outperforms direct labeling of a
(larger) dataset. It also outperforms different learning-from-explanations
techniques trained on the same annotations. Additional experiments

STRENGTHS

- Intuitive approach

- Careful & thorough experimentation

WEAKNESSES

- Writing is difficult to understand in places

- A few details missing needed to interpret numerical comparisons

This paper takes three well-studied techniques (learning from human-generated
feature attributions, learning from noisy labels, and semantic parsing),
and combines them in an interesting and evidently effective way. I think it's a
good fit for NeurIPS.

PRESENTATION

- Given how much there is going on here, I guess it's inevitable that a lot of
  important details wind up in the appendix. Things that really need to go in
  the main paper: (1) identity / compensation of annotators, (2) just a little
  bit of detail about how the semantic parser works.

- Conversely, I think 3.3 can be simplified a lot. Is it really necessary to
  describe everything in terms of modules? If every explanation maps to a
  logical form, then this whole section is (1) if the LF premise applies to a
  sentence, produce its label, (2) otherwise, soften the LF by replacing its
  predicates with more general ones, assign these generalized predicates lower
  scores, and use a probabilistic logic. I think most reviewers will have at
  least passing familiarity with this machinery, and it's not necessary to
  reinvent the wheel.

- This paper has numerous grammatical issues (more than I can list here) and
  would really benefit from careful editing.

- The whole point of the 9th page was to include the societal impact section in
  the main paper; please don't bury it in the appendix if you think it's
  relevant (and if you don't think it's relevant, simply don't include it).

- Some examples and failure cases (even if those have to go in the appendix)
  would be really helpful.


QUANTITATIVE EVALUATION

- I'm not sure how to interpret error bars in tables 3/4/5. Are they standard
  errors of the mean? Standard deviations? 95% CIs? What has actually been
  randomized?  The appendix mentions a seed: does this control model weights
  (i.e.  we're retraining both the upstream and downstream models in each
  experiment) or just the order in which data is presented (as we fine-tune the
  downstream model)?

- What does bolding mean? Some non-bold numbers are identical (e.g. distillation
  / source / HatEval in Tab 4 matches Remote+IG but doesn't get bolded) or
  within error bars (L2 reg / target / StormFront). I suspect that many of these
  differences are not statistically significant, especially after doing a
  multiple hypothesis correction. This certainly isn't a deal-breaker (REMOTE
  seems to be pretty clearly best overall, especially when considering source
  scores and FPRD), but the paper must do some kind of testing (and ideally bold
  all the numbers that are indistinguishable from the best result).


**Time Spent Reviewing:**

1.5

---

> ### Author Response · Authors · 2021-08-10
> **Response to Reviewer fGUd**
>
> Thank you for your positive review and thoughtful comments! Your comments on our paper presentation are very constructive! We will improve the final version based on your feedback and we will definitely polish the writing.
>
> Regarding the quantitative evaluation, we would like to make the following clarification:
>
> ### Q1: How to interpret the error bars in tables 3/4/5?
>
> In Table 3/4/5 and Figure 4, the error bars are showing the standard deviation of the performance when three random seeds are used. The seeds control the order of the data during fine-tuning. During training of the target (downstream) models, the source (upstream) model is a fixed one --- we used the source model that has the best validation F1 score. We will make sure to include more details on the experiment setting in the final version.
>
> ### Q2: What does bolding mean?
>
> The numbers that have the highest average value are bolded. Average value is computed with three runs of model training, each using a different random seed that controls the order of the data during training. We will make sure to include more details on the experiment setting in the final version.
>
> ### Q3: Significance test
>
> To show the statistical significance between our method and baseline method, we do an unpaired t-test between target F1 score on every dataset pairs in our main table between REMOTE and baseline methods. To reduce space, we only include the results for the best baseline method in the following table. It demonstrates that most of the differences are significant.
>
> Significance test between REMOTE and the best performing baseline on BERT-large in Table 3 (Target F1)
>
> -------------------------------------------------------------------------------------------------------------------------------
>
>
> | Dataset Pair         | HatEval -> GHC | Stormfront -> GHC | AmazonMusic -> SST-2 |
> | -------------------- | -------------- | ----------------- | -------------------- |
> | Best baseline method | Distillation   | Distillation      | Distillation         |
> | P-value (Target F1)  | 0.0959         | 0.0073            | 0.0252               |
> | Significant or not   | not quite      | yes               | yes                  |
>
> Significance test between REMOTE and the best performing baseline on RoBERTa-base in Table 3 (Target F1)
>
> -------------------------------------------------------------------------------------------------------------------------------
>
> | Dataset Pair         | HatEval -> GHC | Stormfront -> GHC | AmazonMusic -> SST-2 |
> | -------------------- | -------------- | ----------------- | -------------------- |
> | Best baseline method | L2-reg         | L2-reg            | Distillation         |
> | P-value (Target F1)  | 0.0278         | 0.0012            | 0.3827               |
> | Significant or not   | yes            | yes               | no                   |
>
> -------------------------------------------------------------------------------------------------------------------------------
>
> We will include significant test results for all comparison in our final version.

---

> > ### Comment · Reviewer_fGUd · 2021-08-25
> > **comment**
> >
> > Great, thanks! This review addresses all my concerns and I am leaving my score the same.

---

### Official Review · Reviewer_o8s8 · 2021-07-16

**Rating:** 6
**Confidence:** 4

**Summary:**

This paper presents an idea of using the logical forms of explanations to help refine pre-trained language models.

**Limitations And Societal Impact:**

It would be great to discuss how the proposed ideas can address the inherent biases in pre-trained language models.

**Main Review:**

- First of all, I understand the high-level ideas but had some trouble understanding some technical details. For example, it is not clear to me
  - How to generate refinement advice and how reliable is the generated advice?
  - How to make sure the interaction module and compositionality module can do what we expect them to do? Particularly, for the compositionality module, is it sufficient to use some algebraic operations to replace the logical operations?
  - Which specific algorithm was used to generate the explanation heat-map, given [13] is a survey paper and contains a dozen of algorithms
- Line 169, how reliable the semantic parser is, especially on some out-of-domain data? Will the noisy outputs from the semantic parser have a big impact on the final results?
- The results reported in this work are not directly comparable, as they are evaluated under different conditions. Some ablation studies will be great to demonstrate the benefit of the proposed algorithms.
- Regarding the experiment setup, I don't understand why it makes sense to do evaluations on a sampled subset, instead of directly comparing with "fine-tune(C_all)"?

I would love to update my reviews if all the listed confusion can be addressed.

**Time Spent Reviewing:**

2

---

> ### Author Response · Authors · 2021-08-10
> **Response to Reviewer o8s8 (2)**
>
> ### Q4: Which specific algorithm was used to generate the explanation heat-map?
> Thank you for pointing this out. We used the algorithm from [51] to generate heat-maps in our paper. We will include details about this in the final version.
>
> ### Q5: Line 169, how reliable the semantic parser is, especially on some out-of-domain data? Will the noisy outputs from the semantic parser have a big impact on the final results?
> Besides Sec 3.3, we also include details about semantic parser in Appendix B, including the details about how to evaluate the parser and modify the human explanation. In our experiments, we conduct a round of verification on the solicited explanations, where human annotators can see the output from the semantic parser on the fly. If there is an unparsable explanation, the annotator will either modify the explanation or suggest changes to the parser (e.g., enriching the lexicon). By doing so, we ensure that the parsing will reach 100% accuracy on the collected explanations, over all three datasets (pairs).
>
> To understand how well our parser can generalize to out-of-domain data, we further collected 116 explanations following our annotation guideline, on held-out data from Stormfront->GHC and HatEval->GHC. Note that this set of instances are disjoint from the annotation sets used in our reported experiments. We present the parsing results to human annotators for verification. This evaluation shows that the accuracy of the semantic parser can reach 92.2% on the held-out data (without any update to the parser). Therefore, we believe our semantic parser is reliable when applied to out-of-domain data. We also admit that the parse may encounter unseen vocabulary and typos when human annotators are not strictly following the annotation guideline. We will conduct additional analysis and include more discussion regarding this issue in our final version.
>
> ### Q6: The results reported in this work are not directly comparable, as they are evaluated under different conditions. Some ablation studies will be great to demonstrate the benefit of the proposed algorithms.
> We want to clarify the misunderstanding on the experiment settings used in Table 3 and Table 4.  The two tables are under different experimental settings; while methods within each table are comparable.
>
> In Table 3, we show that within the same human annotation time for collecting labeled data, REMOTE yields better performance than fine-tuning or model transfer methods. The annotation time to provide explanations for REMOTE (which generates R_soft and C_strict) and manually labeling C_sample for the other methods are controlled to be the same (Line 296).
>
> In Table 4, we show the effectiveness of regularization (R_soft) compared with solely using pseudo-labeled data (C_strict or C_soft) as [15] and [45]. Each compared method uses pseudo-labeled C_strict, except that we also provide results of using C_soft for reference.
> To demonstrate the benefit of the main components of our method, we also included ablation studies to compare different choices of each component. In Table 5(b), we show the effect of quantifying interaction for the regularization term. In Appendix E, we compare C_strict with C_soft in Table 9, and compare R_strict and R_soft in Table 10.
>
> We will make the caption and writing more clear in clarifying the experiment settings.
>
> ### Q7: I don't understand why it makes sense to do evaluations on a sampled subset (C_sampled), instead of directly comparing with "fine-tune(C_all)"?
> Thank you for pointing this out and sorry for the confusion caused. We want to clarify that fine-tune (C_all) is an “oracle method” for reference, rather than a valid baseline approach to compete with (as explained on Line 301; Sec 4.1 Compared Methods). To obtain a reference of “upper bound” performance by vanilla fine-tuning, we fine-tune the source model over the fully labeled target-domain data (D_T), which is denoted as C_all. We denote this oracle method as fine-tune (C_all) in Sec 4. Note that C_all are actually unavailable to any of the compared methods (including ours), as these methods only have access to a small subset of instances that are annotated within a limited amount of time (see Line 153 for our method; and Line 296 for obtaining C_sampled). We hope fine-tune (C_all) provides a reference on how much space is left for improvement for different datasets.
>
> We are aware that directly comparing our method’s performance to vanilla fine-tuning methods when trained over the same set of labeled instances is not a fair game, since our method also takes advantage of explanations annotated for these instances (and this process takes additional human annotation time, as discussed in Sec 4.2). Therefore, the comparison with a sampled labeled subset in our experiments is to show that with the same human effort (i.e., the same amount of time spent on data annotation), our method using both labels and compositional explanations yield better performance as compared with methods using only the labeled instance. This is why we control the annotation time of manually labeling target domain data to acquire C_sample (explained in lines 296-301, in Sec ).
>
> We will include more details to make these experiment settings more clear in the final version.

---

> > ### Comment · Reviewer_o8s8 · 2021-08-12
> > **Thank you for the detailed response**
> >
> > The response provides sufficient information to clarify some misunderstandings from my previous comments and help me understand the value of this work. Overall, I think the work is solid.

---

> ### Author Response · Authors · 2021-08-10
> **Response to Reviewer o8s8 (1)**
>
> Thank you for your thoughtful comments and suggestions! We will update the final version based on your feedback and improve the writing. We hope you can reconsider the evaluation of this work based on our clarification below.
>
> ### Q1: How to generate refinement advice?
> First, human annotators use the explanation solicitation interface (Figure 11 in Appendix H) to provide compositional explanations. These solicited explanations contain spurious patterns observed over an individual instance as well as paired refinement advice for adjusting feature attribution and interaction. Their refinement advice would suggest an increase/decrease of attribution/interaction scores of particular phrases. For example, in Table 2, refinement advice is given as, "Attribution score of X should be increased. The label of instance is negative." This refinement advice is then transformed to t_X^{negative}=1, with the notation described in line 247. We will include more details about the explanation solicitation step in the main paper, by moving some content from the appendix.
>
> After the matching process (Sec 3.3), the explanation can be matched to more instances. The matched instance contains a phrase p that matches X, and corresponds to the refinement advice t_p^{negative}=1. The matched refinement advice is then applied to model updating as introduced in Sec 3.4.
>
> ### Q2: How reliable is the generated advice?
> As described in Appendix C (Line 611), three computer science graduate students are involved in the procedure of explanation solicitation. They followed the instructions in Appendix H to provide refinement advice as part of the compositional explanation. Each instance is first presented to one human annotator to solicit explanation, and the collected explanation is further verified by the other two annotators to ensure the annotation quality.
>
> Moreover, to understand the quality of the collected explanations (along with their refinement advice), we included evaluation results on the quality of explanation generalization in Line 327 and Appendix C. We consider any instance’s matching to an explanation "reliable" if the suggested label in the refinement advice aligns with the matched instance’s ground truth label. In Table 7 in Appendix C, we reported precision which is the proportion of matched instances satisfying the above requirement. We can see that strict matching yields more reliable matched instances than soft matching, yielding precision of 0.751/0.717/0.942 for HatEval->GHC / Stormfront->GHC / Amazon->SST-2, respectively.
>
> We will include a more extensive human evaluation on the reliability of refinement advice in our final version.
>
> ### Q3: How to make sure the interaction module and compositionality module can do what we expect them to do? Particularly, for the compositionality module, is it sufficient to use some algebraic operations to replace the logical operations?
> Thanks for this insightful question. The “strict” version of the interaction module exactly follows the parsed explanation to check on the relation between two features described in the B part of the logical rule (B → H) and output a binary label (True or False), as introduced in Line 189 of Sec 3.3. This strict version of the interaction module precisely follows its definition and thus always works as expected. For the “soft” version of the interaction module, we replace the binary output with a numerical score (Line 224-227, Sec 3.3).
>
> Moreover, to understand the effect of “softening” change to the INTER module, we conduct an ablation study on Stormfront->GHC using BERT-base, as shown in the table below. Specifically, we set all other modules in REMOTE as their soft versions but only the interaction module (INTER) as its strict version, and compare it with “REMOTE (all soft)” to show the effectiveness of softening the INTER module. Results show that softening the interaction module is an important operation in generalizing explanations to a broader set of unlabeled instances (as discussed in line 208 in sec 3.3). When we replace the softened version of INTER with its strict counterpart, the performance significantly drops.
>
> Ablation study on the interaction (INTER) module (Stormfront->GHC, BERT-base)
> ------------------------------------------------------------------------------------------------------------------------------
> | Method   				 | source F1    |  target F1 	|  FDRP |
> | ---- | ---- | ---- | ---- |
> | Source model 			 | 57.2 ± 0.7 	| 42.1 ± 1.5	| 16.0  |
> | REMOTE (all soft but INTER)     	 | 51.0 ± 0.9 	| 44.8 ± 0.4     | 15.2	|
> | REMOTE (all soft) 			 | 49.9 ± 3.5 	| 45.7 ± 1.4     | 12.6  | ------------------------------------------------------------------------------------------------------------------------------
>
> The compositionality (COMP) module uses either hard logics (Line 205-207) or soft logics (Line 228-229; Lukasiewiczt logics: "A short introduction to probabilistic soft logic" (Angelika et al.)) to execute the logical operations including “AND” and “OR”. Such design ensures that the module will precisely follow what humans expect them to do. Such soft logic has also been applied in previous work to serve as a differentiable surrogate function for logical operations ("Learning from explanations with neural execution tree" (Wang et al., 2020), "Teaching machine comprehension with compositional explanations" (Ye et al., 2021)). Note that we cannot perform an ablation study for the COMP module. If we set all other modules as the soft version but only COMP as the strict version, the strict COMP module cannot work because the input to COMP will be probability scores (between 0 and 1) instead of binary values (True or False). If we set all other modules as its strict version but only COMP as its soft version, it won’t make any difference because the input to COMP is always binary values.

---

### Official Review · Reviewer_bgag · 2021-07-16

**Rating:** 7
**Confidence:** 4

**Summary:**

The paper deals with the problem of spurious patterns generally learned by a language model. They address this problem by refining the model using compositional explanations. First, they obtain annotation from human workers and then obtain first-order logic from them that map and explain internal features of the model. These updated explanations are used to update the model. They show their results in hate speech and sentiment classification datasets.

**Main Review:**

1. While updating a model with explanations is a neat idea, this reminds me of several works on critique-based explanation generation works from the domain of conversational recommender systems.
2. Quite interesting analysis on annotation time, I wonder how so few explanations could generalize over the full dataset? Is it because of the regularization?
3. Quite solid results otherwise and solid ablations showing the effectiveness of the regularization and how the performance changes with the number of explanations available.

**Time Spent Reviewing:**

2

---

> ### Author Response · Authors · 2021-08-10
> **Response to Reviewer bgag**
>
> Thank you for your positive review and thoughtful comments! We will improve the final version based on your feedback and polish the writing.
>
> ### Q1: This reminds me of several works on critique-based explanation generation works from the domain of conversational recommender systems.
> Thank you for pointing out this line of work! We would like to clarify that our work looks to solicit and model the compositional explanations from human annotators, after presenting them with the post-hoc explanations of the model prediction over individual instances. Rather than studying how to generate insightful explanations like in those critique-based explanation generation works, our work focuses on modeling the compositional nature of the explanations and studies how to generalize explanations to many unlabeled instances such that the explanation regularization can be enforced over a larger set of instances. We believe that there can be interesting integration between our work and the line of work on critique-based explanation generation. We will include a more systematic discussion on this line of related work in our final version.
>
> ### Q2: How so few explanations could generalize over the full dataset?
> Please refer to Table 2, where we describe a concrete example of how one explanation can generalize to unlabeled, with softened matching.
>
> More specifically, we first parse natural language explanations into executable first-order logic rules and then generalize them via three matching modules. These matching modules allow softened constraints so that the coverage of explanations are expanded. Details of the model can be found in Sec. 3.3.

---

> > ### Comment · Reviewer_bgag · 2021-08-31
> > **Thanks!**
> >
> > Thanks for your to-the-point response. I'd like to see the systemic discussion you mentioned in the next version of the paper. I will keep my (already positive) score.

---

### Official Review · Reviewer_CZq1 · 2021-07-17

**Rating:** 7
**Confidence:** 4

**Summary:**

This work introduces a model for incorporating language explanation feedback for domain adaptation with limited labeled data from the target domain. To get this feedback, users are provided with a heat-map of feature attribution scores which they may use to provide feedback on down-weighting / up-weighting attribution scores of certain features, along with labels. Crucially, this paper proposes an approach to convert specific feedback provided on one example, into an abstract template which can then be instantiated on other examples, thus improving coverage of the explanation to many more examples. Finally, this feedback is used to construct a loss which essentially guides the model to up-weight/ down-weight certain feature attribution scores, and this loss along with log-likelihood on the small number of collected labels is used to adapt the model. From experiments, we see that this approach does better than various versions of fine-tuning, hence demonstrating the efficacy of explanations.

**Main Review:**

This is a well written paper with a neat idea of using explanations to finetune models to a new domain. The main insight is that these explanations have “limited regularization strength as it is only enforced on the very few examples associated with human feedback”. However, there are a few things that could be further improved / further elaborated upon:

- What is the role of compositionality in explanations here? Is it just for improving precision in the kinds of examples that are matched? The other place where it could be potentially helpful is in modeling feature interaction, but looking at Table-5(b), it doesn’t seem to help significantly

- In Section 4.2, how are time costs estimated? How many examples is the fine-tune (C_sample) model finetuned on (and how much more data is this compared to C_strict?)

- Looking at Table-3 and Table-4, how is finetuning on C_sample (more data) worse than finetuning on C_strict (lesser data). For e.g. finetuning on C_sample results in 40.7 F1 on HatEval → GHC, and finetuning on C_strict gives 45.1 F1 on the same. Similarly for Stormfront → GHC, we see 49 F1 with C_strict but only 45.0 with C_sample. It appears that finetuning on more data decreases performance.

- Time: How long does it take to finetune with the R_soft loss compared to just straightforward finetuning? It seems like computing the losses in Equation-2 could be slow.

- Hyperparameters: How large is the development set? Typically for domain adaptation, a large labeled development set from the target domain is not practical, and thus hyperparameter tuning is often challenging.

- Related Work: Line 68 claims that REMOTE is the first work to use feature-level supervision from explanations. However, this is also considered in the generalized expectation criteria (Mann et al 2010), where they introduce expectations on the label conditioned on certain features. (e.g. the bigram not bad appearing decreasing probability that a review is negative).

**Time Spent Reviewing:**

3

---

> ### Author Response · Authors · 2021-08-10
> **Response to Reviewer CZq1**
>
> Thank you for your positive review and thoughtful comments! We will improve the final version based on your feedback and polish the writing.
>
> ### Q1:What is the role of compositionality in explanations?
> An improvement in the precision of matched examples is indeed one of the main reasons of using compositional explanations. Also, the compositionality enables annotators to describe their observations more flexibly. This allows them to express more patterns they discovered from the target-domain dataset that may not be able to be described in a single statement. In the example we included in Figure 1, the annotator is able to extract a spurious pattern with "Sweden" and "failure" both included in the explanation. Without compositionality, this inspected example could not provide useful information for model refinement.
>
> ### Q2: In Section 4.2, how are time costs estimated?
> We developed our own annotation interface to support the annotation time estimation. The interface will log the annotation time used in the background. We included screenshots of the interface in Appendix H. We will explicitly refer the readers to Appendix H in Sec 4.2.
>
> ### Q3: How many examples is the fine-tune (C_sample) model fine-tuned on (and how much more data is this compared to C_strict?)
> C_sample has 394/464/204 labeled instances for HatEval->GHC / Stormfront->GHC / Amazon->SST-2, respectively.
> We list the size of C_strict in Table 7 in Appendix C. C_strict has the size of 329/237/1,308 instances for HatEval->GHC / Stormfront->GHC / Amazon->SST-2, respectively.
>
> Please note that C_sample is not always larger than C_strict, because a small number of explanations may strictly match a large number of instances to form C_strict. For example, in the case of “Amazon->SST-2”, 29 explanations can strictly match 1,308 instances in the target domain data D_T, while C_sample has only 204 examples.
> We will include the statistics about C_sample in our final version.
>
> ### Q4: How is finetuning on C_sample (more data) worse than finetuning on C_strict (less data)?
> There are two main factors affecting the performance of the fine-tuning method: the quality of the labeled instances and the quantity of the labeled instances.
>
> Regarding the quantity, we want to note that C_sample is not always larger than C_strict (Sec 4.2). A small number of explanations (annotated within time T) may strictly match to a good number of instances in target domain (D_T), yielding pseudo-labeled data C_strict that has larger size than C_sample. Our response to Q3 includes some statistics for such cases (more can be found in Table 7 in appendix C).
>
> Regarding the quality of labeled instances, it is possible that C_sample has a lower quality than C_strict, yielding a worse performance even though the size is larger. Recall that C_sample is created by randomly sampling from the target domain data (D_T), which does not target misprediction cases. In contrast, C_strict generalizes collected explanations to unlabeled instances (D_T) to create hateful/positive instances, and then randomly samples some unlabeled instances from D_T as non-hateful instances.
>
> Overall, both factors contribute to the final performance of fine-tune (C_strict) and fine-tune (C_sample). We observe fine-tune (C_strict) outperforms fine-tune (C_sample) in some settings and believe that this is due to the mix effect by the two factors mentioned above.
>
> ### Q5: How long does it take to finetune with the R_soft loss compared to just straightforward finetuning?
> The time cost for computing the loss in Equation-2 depends on the proportion of phrases that are given target scores (by generalizing the human-provided explanations). In our implementation, only the phrases that are given target scores will be traversed when computing the loss.
>
> For example, in the sentence "Sweden has proved to be a failure", R_soft is only related to "Sweden" and "failure". Therefore, it is not needed to traverse all phrase pairs to compute the loss for interaction in Equation-2. Typically, very few phrases in a matched sentence would have target scores assigned. For example in the HatEval-->GHC setting, 291 out of the 370 soft-matched instances have only one phrase/phrase pair with target score assigned. The main time cost comes from computing the importance/interaction scores in line 264 and Equation-3. With the hardware and training details specified in Appendix A, it usually takes 50s for one iteration with R_soft loss and 30s without it.
>
>
> ### Q6: How large is the development set?
> Thank you for pointing this out! The size of the source and target domain train/dev/test sets are included in Table 6 of Appendix A. We will include more details about the hyper-parameter setting and selection process in our final version.
>
>
> ### Q7: What is the difference between our work and generalized expectation criteria (Mann et al 2010), where they introduce expectations on the label conditioned on certain features?
> Thank you for pointing this out! Indeed, there is a line of work that associates features with specific labels as a source of supervision, including generalized expectation criteria (Mann et al 2010). Our distinction from this line of work is that annotators are able to provide more complex explanations based on their observations of the generated heat-maps, and we design a matching model to generalize explanations to a wide range of unlabeled instances (as discussed in Secs 3.2 & 3.3). We will include more discussion on this in our final version.

---

> > ### Comment · Reviewer_CZq1 · 2021-08-12
> > **Thanks for the clarifications!**
> >
> > Thanks for clarifying finer details about C_sample and C_strict! I had some follow ups:
> >
> > 1. **Could you elaborate a bit more on how C_strict can have a higher quality than C_sample?**
> >
> > Your answer would apply if the explanations are solicited  *only for mis-classified instances* and hence the pseudo-labeled examples in C_strict would then explicitly target the misclassified slices from the target distribution.
> >
> > But from my understanding, the explanations are obtained on randomly sampled data from the target domain. As a result, the pseudo-labels will not be differentially applied on misclassified slices. Thus, C_strict would not necessarily consist of a skewed distribution targeting particular parts of the target domain that the model struggles on.
> >
> > On the other hand, if the pseudo-labels were obtained only for misclassified instances, then I buy the argument that C_strict contains “higher quality” examples. (would be nice to do this control experiment if this is indeed the reason for why C_strict is a better sampling than C_sample)
> >
> > 2. **For Stormfront $\rightarrow$ GHC and HatEval $\rightarrow$ GHC, C_sample is actually larger than C_strict (in-fact almost twice the size for HatEval $\rightarrow$ GHC) . So, why is it so much worse to finetune on C_strict vs C_sample?**
> >
> >
> > IMO, most of the gains can be attributed to the difference between C_sample and C_strict (looking at Table-4 and comparing Target F1 of row-2 and row-6/7, R_soft has a smaller boost) , so it would be good to spend a larger chunk discussing this.

---

> > > ### Author Response · Authors · 2021-08-26
> > > **Response to follow up questions from Reviewer CZq1**
> > >
> > > ### 1. Could you elaborate a bit more on how C_strict can have a higher quality than C_sample?
> > >
> > > Thank you for the follow-up question. We would like to clarify that human annotators are presented with _randomly_ sampled data from the target domain (D_T). They inspect the heat-maps of _all_ these instances, but explanations are _only_ created when human annotators identify _spurious patterns_ on these instances (Line 156-157; “write an explanation if a spurious pattern is found”). We reported the yield rate of explanations for this process in Line 314 of Sec 4.2: around 15% for hate speech detection (Stormfront -> GHC and HatEval -> GHC) and 62% for sentiment analysis (Amazon -> SST2), respectively.
> > >
> > > We do not limit the explanations to be associated with only mispredicted instances. The collected explanations can be associated with either correctly or wrongfully predicted instances. Empirically, we find 76.5% of collected explanations to be associated with mispredicted instances in the HatEval->GHC setting. In this case, C_strict has a higher quality than C_sample in terms of refining the model from its spurious patterns and mispredictions. . We will include more in-depth analysis on the changes of model’s behaviors in our final version.
> > >
> > > Regarding the quantity, we are sorry for the confusion caused by our earlier response.  Note that for hate speech detection tasks, we randomly sample instances from the target domain (D_T) and treat them as non-hateful examples. This is to avoid train-test label distribution mismatch, because C_strict contains mainly hateful instances. (Line 316-321, Line 380-385). For example, in the Stormfront->GHC setting, the effective size of C_strict is 237 (matched instances) plus 1600 (sampled instances that are treated as negative examples; detailed statistics can be found in line 647-656 in Appendix E). So overall the instances we use for C_strict fine-tuning (237+1600=1837) is more than C_sample (464).  For sentiment analysis task we only use matched instances for experiments so the numbers remain the same. We will clarify this and include a more detailed description in our final version.
> > >
> > > ### 2. IMO, most of the gains can be attributed to the difference between C_sample and C_strict, so it would be good to spend a larger chunk discussing this.
> > >
> > > We agree with the reviewer and will include more detailed discussion on the quantity and quality of C_strict versus C_sampled in the final version.

---

### Author Response · Authors · 2021-08-10
**General response to all reviewers**

We thank the reviewers for their positive reviews. We are delighted that the reviewers found our work to be novel, well-motivated, intuitive, and supported with extensive experiments. We very much appreciate the helpful comments regarding improving the clarity of our presentation, including additional analysis on some parts of the proposed method, and elaborating on implementation details. We will carefully address these comments in our final version, by moving some content from the appendix, including additional analysis, and improving the writing.

We respond to the individual reviewers’ comments in details below.

---

### Decision · Program_Chairs · 2021-09-27

**Decision:**

Accept (Spotlight)

**Comment:**

The paper introduces an approach to incorporating natural language explanations into model fine-tuning. Annotators are presented with a heat-map of feature attribution scores on a target unlabeled corpus and they are asked to refine the importance scores and describe interaction of features in language. Description are then parsed into logical rules, which provide abstract templates that can be instantiated on other examples, thus improving coverage. Then the model is then regularized to align with the pseudo-labeled attribution data.

--

The paper tackles the important problem of efficiently exploiting costly human explanation on few examples by generalizing such explanations "compositionally" to other examples: to do so, the method effectively uses known techniques in semantic parsing, pseudo-labeling and feature attribution methods. All reviewers and myself agree that the method is interesting and novel, the experimentation is "careful & thorough" and the results confirm the effectiveness of the pseudo-labeling approach. I suggest the authors to incorporate all the precious reviewers' feedback. Following reviewers' feedback, extra care should be put into proofreading the paper for grammar and spelling errors (some I can think of right now are in Fig. 2 caption "annotaters", "rulesand", ...).

I recommend this paper for acceptance.